# Divergent convective outflow in large eddy simulations

Edward Groot[1] and Holger Tost[1]

[1]Institut für Physik der Atmosphäre, Johannes Gutenberg Universität, Johannes-Joachim-Becher-Weg 21, Mainz, Germany

**Correspondence:** Edward Groot (egroot@uni-mainz.de)

**Abstract.** Upper tropospheric outflow is analysed in cloud resolving large eddy simulations. Thereby, the role of convective organisation, latent heating and other factors in upper tropospheric divergent outflow variability from deep convection is diagnosed using a set of more than 80 large eddy simulations, because the outflows are thought to be an important feedback from (organised) to large scale atmospheric flows: perturbations in those outflows may sometimes propagate into larger scale perturbations.

Upper tropospheric divergence is found to be controlled by net latent heating and convective organisation. At low precipitation rates isolated convective cells have a stronger mass divergence than squall lines. The squall line divergence is the weakest (relative to the net latent heating) when the outflow is purely 2D, in case of an infinite-length squall line. At high precipitation rates the mass divergence discrepancy between the various modes of convection reduces. Hence, overall the magnitude of divergent outflow is explained by the latent heating and the dimensionality of the outflow, which together create a non-linear relation.

## 1  Introduction

Organised deep moist convection is not only a substantial precipitation source over the tropics and mid-latitudes, but also a driver of the global atmospheric circulation due to its conversion of potential energy into kinetic and moist static energy. This energy conversion is achieved by so-called latent heating: condensation of water vapor warms rising air parcels while they move upward, expand, and cool. The warming tendency of latent heating opposes the stronger cooling tendency (expansion) and provides (positive) buoyancy. The positive buoyancy is the "fuel" to the moist convection that can keep it running, even accumulate, and further organize.

Once organized systems of deep moist convection (from now on "convective systems") have formed, they feed back onto the background atmospheric circulation. The background atmospheric circulation is hardly affected by tiny convective systems composed of one or two cumulonimbus clouds. On the other hand, the mesoscale circulation can be entirely disturbed and even dominated by convective systems of sufficiently large size and intensity (Houze, 2004, 2018): in case of so-called mesoscale convective systems (MCS) a complete re-organisation of the atmospheric flow around the MCS can happen. In other words: large systems with higher precipitation rates introduce an on average stronger feedback to the large scale atmospheric flow (intuitively), i.e., the feedback is expected to increase with the precipitation intensity (or equivalently net latent heating). Consequently, the net latent heating can be used to quantify the intensity of convective systems. The feedback onto the background circulation has sometimes also significant consequences for downstream developments of the atmospheric flow (e.g. Rodwell

et al., 2013; Clarke et al., 2019a, b).

An increase of the flow feedback strength with the amount of net latent heating is supported by the simplified linearised gravity wave model described in Bretherton and Smolarkiewicz (1989), Nicholls et al. (1991), Pandya et al. (1993), and Mapes
(1993). Principles behind this linear gravity wave modelling approach have additionally been used for simulations of the flow feedback from squall lines by Pandya and Durran (1996). Furthermore, the model has also been used in a very different set-up to study flow adjustments to localised heating by Bierdel et al. (2018). Bretherton and Smolarkiewicz (1989) studied gravity wave responses to such heat sources with a linearised model that supports gravity waves. Their linear model reveals increasing convectively induced circulation with an increasing latent heat source. The gravity wave adjustment signal propagates away
from the convective system, but comparatively strong upper tropospheric outflow is maintained around the location of the initialised latent heat pulse. Moreover, their model describes how point and line sources generate different outflow responses and how responses depend on vertical wavenumber: in other words, the outflow may depend on the organisation of the convection (including storm geometry). Extensions of the linear model by Bretherton and Smolarkiewicz (1989) have later been used to understand preferential locations of convective initiation, e.g. in the tropics (Lane and Reeder, 2001; Stechmann and Majda,
2009; Lane and Zhang, 2011; Grant et al., 2018), and to understand error propagation associated with differential heating in a rotational set-up (Bierdel et al., 2017). The linearised model by Bretherton and Smolarkiewicz (1989) can serve as a benchmark for the irrotational cloud resolving and large eddy simulations with much more complexity presented here (for most important predictions of the gravity wave model, here partly used as assumptions, see Bretherton and Smolarkiewicz, 1989; Bierdel et al., 2017, 2018).

Over the last decade, studies on predictability have suggested that differential upper tropospheric flow variability induced by organised convective systems can impact predictability of weather (e.g. Rodwell et al., 2013; Baumgart et al., 2019). Hence, the main objective of this study is to understand and diagnose the upper tropospheric outflow variability and feedback of organised convective systems to its surroundings quantitatively for large eddy simulations. More specifically, the upper tropospheric mean lateral acceleration over a control volume, as diagnosed with the mass divergence, is connected to the net latent heating.
Analysing a control simulation, an ensemble, and tailored physically perturbed experimental simulations, this study systematically assesses the effect of net latent heating, convective momentum transport and convective organisation on the divergent outflow of convective systems. The divergence sets on as a horizontal wind compensation for vertical acceleration of convective upward airflows, accompanied by a high pressure anomaly aloft. Such a high pressure anomaly aloft exists as a consequence of abundant mass aloft.

Ensembles and physical perturbations are applied to selected organisational modes of convection: a supercell, regular multicells, and a squall line. The latter class is further sub-divided into two categories (finite-length and infinite-length squall lines). Convective momentum transport is purposely switched off or adjusted by $\pm 50\%$. Additional physical perturbations are applied to the aforementioned four basic modes of convection (scenarios) to test specific hypotheses and to improve the quantification of the impact of latent heating.

Large eddy simulations are suitable for the assessment, because of explicitly resolved turbulence and cloud-scale processes represented in those simulations. Therefore, they can assumed to represent the convective processes in a reliable way (e.g.

Bryan et al., 2003).

The structure of this manuscript is as follows: in Section 2 the model set-up (Section 2.1), initial conditions for four prototypes of convection, and corresponding convective environments (both in Section 2.2) are described. Furthermore, all perturbation types (including ensemble configuration) are covered in this Section (Section 2.3) and the analysis window is described (Section 2.4). In Section 3 the evolution of convective cells is first discussed for the reference simulations. Thereby, each of the four prototypes of convection are introduced separately (Section 3.1). This part is followed by an analysis of the vertical motion caused by the convective adjustment. The next part discusses the internal variability in the investigated set-up using the ensemble (Section 3.3). Furthermore, suitable vertical masks of the convergence and divergence are investigated in that Section. After defining all the constraints, a dataset with integrated outflow divergence patterns can be created in Section 3.4. That dataset sheds light onto the relationship between latent heating and upper tropospheric divergence. The manuscript is finalised with a discussion and a conclusion section.

## 2 Methods

### 2.1 Model set-up

Simulations presented in this study are conducted with cloud resolving model CM1, version 19.8 (Bryan, 2019). The default horizontal grid size is 120 by 120 km, with a default simulation time of 2 hours ($9600dt$). The vertical extent of the domain is 20 km. A sponge layer exists in the upper 5 km, which damps upward propagating gravity waves. Output is stored per 5 minute interval. The simulations are run in large eddy simulation (LES) mode at $dx, dy = 200$ m and $dz = 100$ m by default. In addition, extra simulations are run with additional grids where $dx, dy = 100$ m, $dx, dy = 500$ m and $dx, dy = 1$km. In the latter two the vertical grids are adjusted to 250 m and 500 m intervals. In one last simulation with adjusted resolution, $dz$ is set to 200 m.

In LES-mode, a TKE-scheme after Deardorff (1980) handles the subgrid turbulence. For the microphysics, the default CM1 scheme is used: the two-moment scheme of Morrison et al. (2009). Boundary conditions of the simulations are of the non-periodic, open type. Hence, derivatives of any quantity are set to 0 at the boundaries in every direction: an infinite reservoir of inflow air is theoretically available to the convective systems. As a consequence, reflection of wave signals can also occur at the boundaries. For more details on the model settings (dynamics, physics) we also refer to Groot and Tost (2023).

### 2.2 Environmental conditions

The initial thermodynamics is prescribed using the profile of Weisman and Klemp (1982) (Appendix A, Figure A1), a standard in CM1. Two basic local potential temperature perturbations have been set at $t = 0$ to trigger convective cells with various kinds of organisation. Furthermore, the initial wind profiles are varied at $t = 0$ to realise systems that manifest with a certain organisational mode of convection, numbered # 1-3 (see left side of Figure A1 in the Appendix; Rotunno et al. (1988); Weisman and Klemp (1982); Bryan (2019) and Bryan's CM1 code). Each of the four combinations of local temperature perturbations

and wind profiles are introduced below. These prescribed profiles establish the four modes of convection. An overview of all the scenarios introduced in this section is shown in Figure 1.

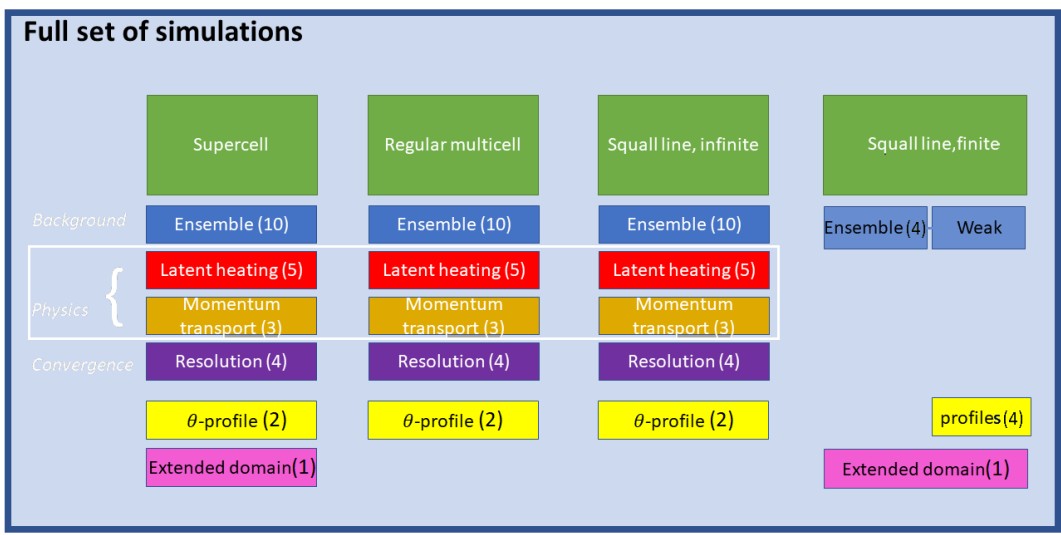

**Figure 1.** Overview of all CM1 experiments done in this study. The four scenarios represented in the columns of the display have been introduced in Section 2.2. These green boxes with white caption show each of the four prototypes of convection that we use, with a list of experiment groups following in the column below each of them. Below the column header, the perturbations applied to each scenario are represented, which are discussed in the order of display (downward; Section 2.3). Here, white text represents the regular basis set of perturbation experiments applied to the first three modes of convection and black fonts represent irregularities among the experiments, tailored at specific modes of convection and robustness testing. In the last column "weak" denotes the ensemble band corresponding to the other three scenarios and "profiles" denote the wide ensemble band, which correspond to $\theta$-profile perturbations in the other three scenarios (Sections 2.3.1 and 2.3.3).

### 2.2.1 Supercell

A supercell scenario is constructed by applying an initial warm bubble disturbance. The warm bubble is initialised around the center of the domain with a radius of 10 km in the horizontal. It has a bell shaped amplitude with a maximum of 1 K at the origin and $z = 1.4$ km. The warm bubble forces upward motion in the domain center, which in combination with the high convective available potential energy (CAPE) leads to the growth of strong convective cells.

A strong wind shear profile (#1, see Appendix A, Figure A1) induces a supercell structure. $u$ gradually increases from easterly winds of 12.5 m/s at the surface to 18.5 m/s from the west at interface between the layer of wind shear and that of unsheared winds, located at $z = 6000$ m in the reference case (Weisman and Klemp, 1982). This combination of strong easterly and westerly wind kept the convective system centered within the domain throughout its evolution. The $v$ wind varied from -2 m/s at the surface to +2 m/s at and above that same interface.

### 2.2.2 Regular multicell

A regular multicell is generated in the same warm bubble initiation scenario as the supercell case.

Moderate wind shear is applied, in combination with the warm bubble scenario, leading to ordinary regular multicell convection. However, the easterly inflow at the surface is set to 11 m/s, while $u$ increases to +3 m/s at the interface height located at 2.5 km altitude (adjusted from Rotunno et al., 1988, ; #2, Appendix A, Figure A1). Again, the wind profile is designed to keep the convective system relatively centered within the domain throughout its evolution.

### 2.2.3 Infinite-length squall lines

An infinite-length squall line is constructed with a cold pool damming scenario, in which west of the y-axis a -6 K potential temperature perturbation is implemented at the surface. This perturbation decreases linearly with height to 0 at a fixed level of $z = 2.5$ km. Upward motion that initialises convection is generated at the border between the air masses in the west and the east. The combination of high CAPE and the moderate shear profile perpendicular to this border leads to a strong line of convective cells.

A moderate wind shear is applied, similar to the regular multicell scenario. With this moderate shear, the u-component of the wind is linearly varying from 12.5 m/s of easterly inflow at the surface to weak westerly flow of 1.5 m/s above the top of the shear layer (Figure A1), which is set at $z_{i,ref} = 2500$ m for this scenario in the reference run (adjusted from Rotunno et al., 1988, ; #3, Appendix A, Figure A1). As in the other two modes of convection, $v$ increases from -2 to +2 m/s over the 2500m deep low level shear layer.

### 2.2.4 Finite squall line simulations

Despite the substantial similarity to the infinite squall line, this additional class of organised convective systems is constructed to obtain convective cells arranged in a line, but with more potential for outflow in the y-dimension locally (at least at the squall line edges). This meridional outflow is substantially damped in an infinite-length squall line. Most of the conditions are identical to the infinite-length squall line simulations (described above). The main difference is a modification of the initial potential temperature perturbation in the cold pool damming scenario: in the central region of the domain, it maximises at 6K at the surface, but this surface maximum decreases outward to 0K near the meridional boundaries of the domain (Figure A2 in Appendix A illustrates the $y$-$z$ shape of the profile). Quantities of interest are separately diagnosed over a central region (comparable to the infinite squall line) and an outer region in the finite-length squall lines (on both the northern and southern ends) for this scenario.

### 2.3 Perturbations

#### 2.3.1 Ensemble perturbations

To test the robustness of the results an ensemble is constructed in the following way, and for each scenario accordingly: the altitude of the interface between the layers where shear is initially present and absent is perturbed (symbol $z_i$). Thereby, the maximum deviation within the ensemble is 5% from the reference $z_{i,ref}$. Corresponding extreme deviation values for $z_i$ are $2500 - 127$ resp. $6000 - 304$ m, with an ensemble mean deviation of 2.7% from the reference altitude. Relative magnitudes of the perturbations are equal for both shear profiles, and hence, all storm modes.

Ensemble perturbations for the finite-length squall lines (Section 2.2.4) are set-up in a slightly adjusted way compared to the other three scenarios: four perturbations are generated as a "narrow ensemble band" where the depth of the initial shear layer is adjusted, similarly as for the infinite squall line. On top of that, four additional ensemble members are generated with stronger initial condition perturbations: one with a 1 km deeper cold pool, one with a deeper (climatologically more realistic) shear layer, and two with a 4K and 7.5K maximum of the potential temperature perturbation in the domain center at the surface, referred to as a "wide ensemble band".

Ensemble perturbations provide a background scatter for the natural variability of upper tropospheric divergence within a close proximity of the control runs, as caused by small variations in initial conditions. After applying interface perturbations, winds are interpolated to the native vertical grid length of the corresponding simulation: 100 m.

#### 2.3.2 Physics perturbations

Two types of physics perturbations are applied. These perturbations are applied for a comparison to the control simulation of each of the three basic modes of convection (Sections 2.2.1, 2.2.2 and 2.2.3, three left columns of Figure 1).

The constant of latent heating is adjusted to 60%, 80%, 90%, 110%, and 120% of its actual value. This approach has been selected to serve as a proxy for perturbed cloud and microphysics tendencies (e.g. condensation, evaporation) or CAPE, without perturbing any of the other physics and the initial environment within the model. Precipitation rates in simulations with an adjusted latent heating constant are naturally evaluated with the latent heating constant adjusted correspondingly in any conversion.

The vertical advection term in both horizontal momentum equations has been adjusted to 0, 50%, and 150% of its actual magnitude in another set of experiments, to perturb the convective momentum transport. This is done to determine the direct effect of the convective momentum transport process on upper tropospheric divergent outflows. The perturbation is similar to creating an artificial source/sink term of horizontal momentum at locations with strong convective motion, which is driven by tendencies caused by vertical gradients in horizontal momentum.

Systematic non-linear effects on the mass divergence are detectable, if processes other than the intensity of the convection affect mass divergence. Additionally, the role of other parameters such as convective organisation and convective momentum transport for the upper tropospheric divergence can be determined by the comparisons between simulations.

### 2.3.3 Adjusted low level temperature perturbations

The dataset obtained with simulations introduced in Section 2.2 is complemented with additional simulations, in which the strength of the potential temperature disturbances (warm bubble(s) or cool pool damming) has been adjusted. These modifications result in slightly stronger (weaker) triggering and hence slightly stronger (weaker) convective cells would be expected compared to the other ensemble simulations. The configuration is as follows: the initial perturbations were halved or otherwise slightly modified, using scaled superpositions of the cold pool and warm bubble initiations.

### 2.3.4 Simulations at extended domains

The domain size that has been chosen in this study is on the small end for studying the feedback from convective cells to their environment: especially for the supercells and squall lines and during the last half an hour of the simulations. In the regular multicell simulations, however, the limited domain size should be of no concern in this regard: the convective cells cover only a limited fraction of the 120x120 km domain.

To test the effective limitations of the restricted domain and more robustly determine the patterns in our dataset (and herewith strengthen the conclusions), one supercell and one finite-length squall line simulation are conducted at an extended domain (200 by 200 km). The simulation time is extended to 160 minutes, but the analysis window is restricted to the two time intervals until 120 minutes. For the finite squall line, the large domain simulation configuration is not identical to the reference squall line simulation, but uses the conditions for an ensemble member with reduced potential temperature perturbations: maximum 4K only. This configuration is selected to prevent too much additional convective initiation (secondary) with convective precipitation. Such secondary convective initiation is partially located further away from the squall line. The additional convective initiation makes the evolution of the system less comparable to the ensemble simulations in the reference domain.

### 2.4 Spatial and temporal analysis windows

Diagnostics that represent latent heating by precipitation, upper divergent outflow and convective momentum transport are evaluated over two separate time intervals. The first time interval ends after 75 minutes for the squall lines and 90 minutes for the regular multicells and supercells. Diagnostics are also evaluated over the second time interval, running from the end of the first interval until the end of the simulation (120 minutes). This approach with two time intervals creates temporal subsamples. In the first interval's subsamples, effects close to the selected box boundary are relatively unimportant . On the other hand, such effects have a comparatively stronger impact on the diagnostics during the second time interval. Comparison between the two intervals helps to determine the relevance of, for instance, propagation of gravity waves influencing the larger scale environment.

Furthermore, a restricted rectangular horizontal area within the whole domain is selected, over which diagnostics are averaged spatially, further limit boundary effects. The exact extent of the boxes is depicted in the respective Section 3.2, that follows.

 **3  Results**

In this section the development of the convective systems is described from an introductory point of view, by illustrating the simulated reflectivity and describing the evolution of the precipitation systems (Section 3.1). This is done for the control simulation of each of the four modes of convection separately. Once the horizontal distribution of the convective heat sources and region of flow adjustment is known (Sections 3.1 and 3.2; see also Figure S1 and Section 1 in the supplement), the last

ingredient needed for the box analysis and diagnosis of the upper tropospheric mass divergence from the convective systems is delineating the vertical extent over which the divergent outflows develop (Section 3.3). Finally, this section is concluded with the dataset of diagnosed mass divergence and net latent heating for all simulations and both time intervals, where the mass divergence is based on the horizontal and vertical extent of the box (Section 3.4). That dataset bridges the gap to the discussion that follows.

**3.1  Evolution of the convective cells**

Figure 2 depicts the temporal evolution of the four convective systems in the control simulation, together with their corresponding simulated patterns of radar reflectivity.

**3.1.1  Supercell**

The initial warm bubble is a source of buoyant air around the origin, which can freely ascend. Part of it develops into a deep

convective cell and in the conditions of high shear, it organises itself as supercell. After 25 minutes, the cell develops and simulated radar echoes appear at 3 km altitude (Figure 2, left column). The cell stretches out strongly in the east-west direction under the condition of a deep layer shear larger than 30 m/s. A hook echo appears after about 45 minutes in a southern cell, about 10 km west of the origin, with an antisymmetric cell as northern counterpart. The southern hook echo starts accelerating southeastward and thereby still gains size. On the western flank, initiation of much smaller convective cells sets on after about

85 minutes.

**3.1.2  Regular multicell**

From the warm bubble initialised at the origin, a convective cell is able to develop right next to the origin as in the supercell simulations (Figure 2, second column). This is a consequence of weak upper level flow and strong surface inflow with high CAPE values as given by the Weisman and Klemp (1982) initial conditions and a buoyant warm bubble.

After 25 minutes of simulation the first echo signals appear at $z = 3$ km, directly below melting level. A first convective cell remains small during the first hour, with size of 10 by 20 km in the horizontal direction and maximum reflectivity around 60-65 dBz. A small cold pool develops on the downdraught side (west). During the next output time steps, the precipitation system remains contiguous, but also develops two cores (around and just after 60 minutes): a southerly and a northerly cell. Herewith a two-cell system, a multicell, has developed.

### 3.1.3 Infinite-length squall line

In the infinite-length squall line simulations deep convection develops along the cold pool edge, which sits at the y-axis (Figure 2, third column). Convective initiation occurs as upward motion is triggered at the interface between warm air to the east and cool near-surface air in the west. With substantial amounts of CAPE, shear helps to organise the convective storms along the y-axis.

The first precipitation cells appear along the y-axis after 15 minutes. A secondary phase of convective initiation occurs a few kilometers ahead of the main squall line after 30-40 minutes of simulation time, which is more extensively discussed in Groot and Tost (2023). Newly initiated convective storms exceed reflectivities of 55 dBz, with values up to about 65 dBz locally. This is followed by an onset of eastward displacement of the line of convective cells.

The evolution of the squall line ensemble spread is discussed very extensively in Groot and Tost (2023). The key finding is that the essential developments for the ensemble spread occur with the secondary convective initiation, with subsequent differences in cold pool acceleration within the ensemble.

### 3.1.4 Finite squall line

The finite squall line starts precipitating after 15 minutes over a length of about 50 km along the y-axis (Figure 2, last column). After 20 minutes, reflectivities above 65 dBz already appear in the model output and the precipitating region grows in each horizontal direction. Cellular structures are not yet present, but start appearing after 30 minutes of simulation time. By this time its length is about 75 km, centered at the origin.

While the core region maintains its position near the centre of the domain, an extension of the squall line at both ends ($y \approx \pm 40$ km) adjusts the geometry of the convective system to an arch-shaped line after 60-65 minutes. Simultaneously, some convective initiation occurs locally, west of $x = -40$ km at $y = \pm 40$ km. These small cells live for maximum 5-10 minutes. The associated precipitation accumulation is negligible compared to the rest of the squall line (initial conditions were selected to reduce the size and duration of cells as much as possible on purpose).

With the development of the arching geometry, the simulated reflectivity signal strengthens and the convectively active area in the outer regions (close to the northern and southern boundary) increases as well. The squall line center region starts to accelerate eastward and moves to about 15-20 km east of the origin over the last 40 minutes of the simulation. However, this acceleration is mostly restricted to a 30 km region around $y = 10$km.

**Cumulative precipitation**

The precipitation cells in all four modes of convection do not move far from their original position near the origin, as displayed in each of the panels of Figure S1 in the supplementary material. As the divergent outflows could reasonably be assumed to be collocated with cumulonimbus clouds (the regions of diabatic heating) and their close proximity and thus with the precipitation signal, net outflow has to (mostly) stick to that region near the central part of the domain. That suggests that an integration over a subdomain of the simulation domain suffices for rigorous assessment of outflow magnitudes in the simulation dataset.

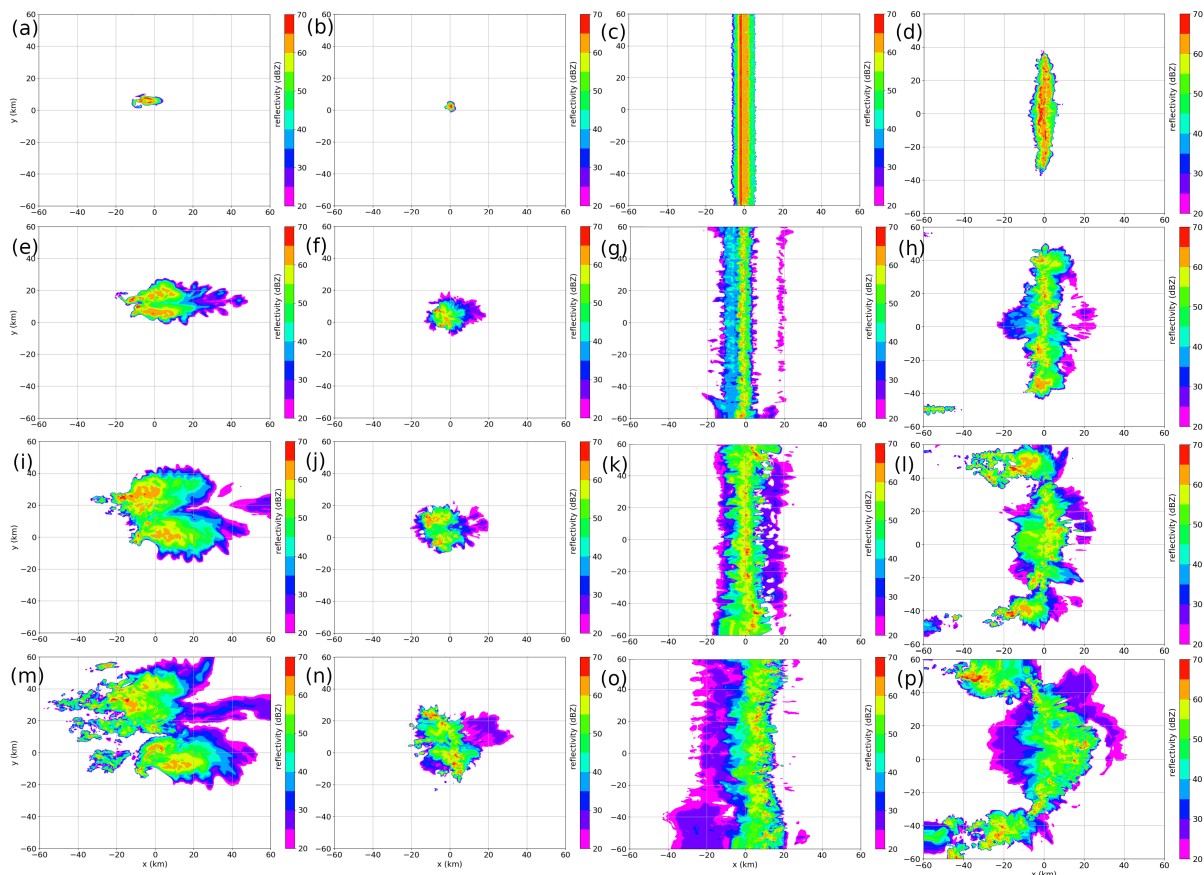

**Figure 2.** Simulated radar reflectivity at 3 km height in the control simulation for each of the four modes of convection: from left to right Supercell (a, e, i, m), regular multicell (b, f, j, n), infinite-length squall line (c, g, k, o) and finite-length squall line (d, h, l, p). From top to bottom time increases: 30 min (top row, a, b, c, d), 60 min (second row, e, f, g, h), 90 min (third row, i, j, k, l) and 120 min (bottom row, m, n, o, p).

## 3.2 Vertical motion

Figure 3 shows the vertical velocity at a level near the tropopause (within 0.5 km). The boxes over which further diagnostic quantities are integrated are also outlined accordingly as black rectangles. The simulations are split into two time intervals, as mentioned in Section 2.4: the first interval before the snapshot in Figure 3 and the second interval, covering the part of the simulation afterwards. The box is chosen such that the flow effect of the convection through rippling in the wave signal at the tropopause level is still limited to the region (mostly) within that horizontal extent. By the time of the snapshot in the figure, only a fraction of the longwave gravity wave signal left the region of the box.

Comparing the four modes of convection, Figure 3 shows clear contrasts. The supercell simulation (left) has a large region over which shortwave signals occur near the tropopause in comparison to the regular multicell. The box size over which outflow

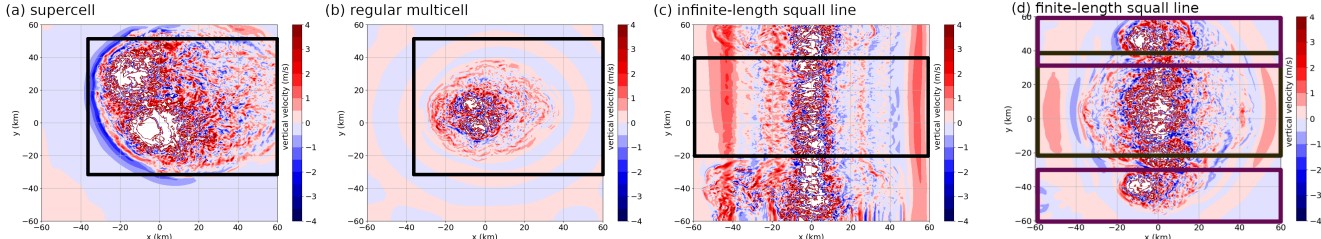

**Figure 3.** Vertical velocity at tropopause level after 90 minutes for the control simulations of (left flank) the supercell simulation, (center left) regular multicell and 75 minutes (center right) for the infinite-length squall line and finite-length squall line (right flank). The thick black outline of the rectangle defines the outer region of the horizontal box over which diagnostics are integrated and the time stamp that belongs to it defines the end point of the first integration interval. That stamp corresponds the start of the second integration interval as well. Areas that mostly exceed $\pm 4$ m/s indicate direct convective motion, which means that an updraught (downdraught) core is present locally.

is diagnosed has to be sufficiently large that it covers the adjustment region of shortwave gravity wave activity. On the other hand, shortwave variability only occurs in an oval that is restricted to a comparatively smaller region in the regular multicell simulation (second panel). For a fair comparison, the supercell and regular multicell integration boxes are set over the same spatial extent, relative to the initial warm bubble. A region where near-tropopause boundary reflection effects (as a result of the open boundary conditions) are suspected to occur can be identified in the infinite-length squall line simulation: regions away from the centre ($y = 0$) have a wider zonal extent (about $x > \pm 35$ km) over which shortwave $w$ variability is strong (namely at $y < -20$ and $y > 40$) compared to the central region. That pattern in $w$ is not present in the display of the finite-length squall line as a consequence of the arch shape, which only covers a restricted part of the domain. The patterns of vertical motion as a consequence of gravity waves and convection occurring in the middle troposphere are discussed in further detail in Section 2 of the supplement. This is the level where the wave amplitude of the fastest mode of gravity waves maximises.

### 3.3 Divergence profiles

Figure 4 shows the time evolution (x-axis) of the vertical divergence profiles (y-axis) for each of the four basic types of convective organisation (starting from $t = 5$ minutes). Initially, the convection has not developed. It requires about 30 minutes before intense convection develops, as it can be visualised with the selected threshold and corresponding color scale in the figure. Note, that the color scale of the isosurfaces and the isolines use different values to allow for a distinction between the pattern of individual ensemble members and the ensemble mean value of strong divergence. In the top and bottom row strong convergent low-level ($z < 3$ km) signals start to exceed the isoline threshold after about 45 minutes ($\pm 20$ minutes). With a slight delay (about 15-20 min), strong signals of mass divergence set on and stick to a layer between 8 and 13 km, around and just below the tropopause.

The mass divergence (convergence) signals in the regular multicell both set on after an hour in the upper (lower) troposphere. Afterward they expand their vertical extent with time. In this set of simulations the upper tropospheric divergence also sticks

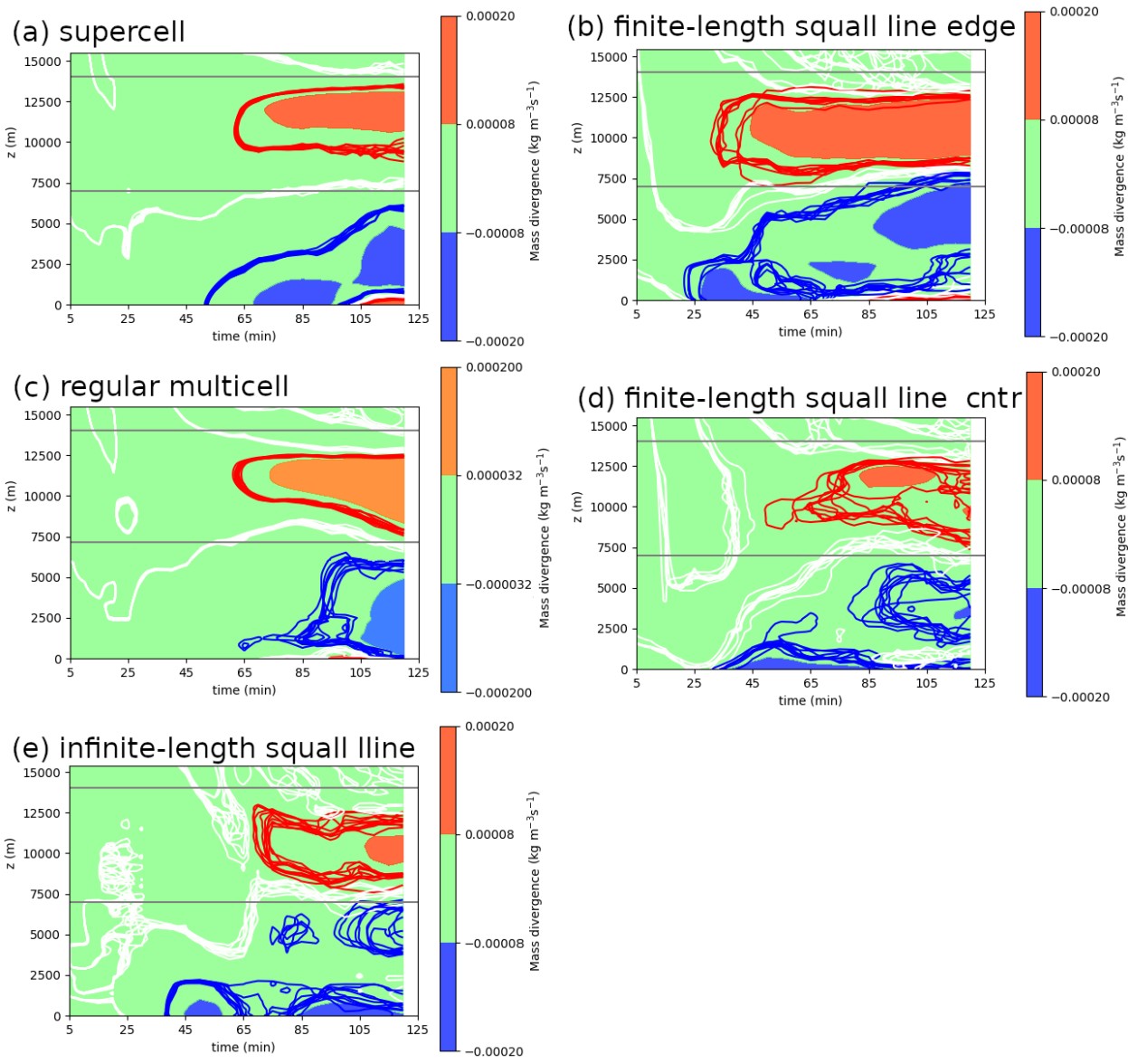

**Figure 4.** Time evolution of mass divergence (convergence) as a function of height for three basic modes of convection, averaged over the ensemble (filled) and for the individual members (spaghetti contours; blue: $-5 \times 10^{-5}$ kg m$^{-3}$s$^{-1}$, white: 0 kg m$^{-3}$s$^{-1}$ (i.e. neutral divergence/convergence), red $+5 \times 10^{-5}$ kg m$^{-3}$s$^{-1}$ (four of the five plots) and $\pm 2 \times 10^{-5}$ kg m$^{-3}$s$^{-1}$ (left column, "cntr", i.e. center)). Note that the contouring values differ from those corresponding to the color fill. The finite-length squall line is further split-up into its edge and center regions (right column).

to the 8-13 km layer. The low level convergence expands quite much after about 1.5 hours, as to nearly cover the full lower half of the troposphere.

The ensemble spread is narrow for Figure 4 a and b, as suggested by the spaghetti lines. Among the infinite-length squall line ensemble members there is more spread: the lower panel suggests a typical spread up to about 1-2 km in the vertical for each contour of mass divergence (convergence) during the second hour (maximum/minimum level of isoline). This is confirmed by numerical analysis: the time average of the layer depth over which $\mu_{ens} \pm \sigma_{ens}$ overlaps with zero divergence, from $t = 45$ minutes until $t = 120$ minutes and over the troposphere (2-12 km altitude), is 300-350 m for the regular multicell and the supercell. Conversely, for the infinite-length squall line the value is 975 m. Groot and Tost (2023) provides a detailed analysis of the ensemble evolution in this set of simulations.

Of particular interest is the mid-tropospheric contour of neutral convergence in Figure 4 (white isolines), as this marks the boundary between the upper tropospheric divergent outflow and the entrainment/inflow region of convection in each simulation. It settles at about 5-6 km altitude, after rather noisy behavior in the first 30 minutes due to (mostly) undeveloped convection. It rises to 7-8 km altitude for each mode of convection eventually (after about 90 minutes of simulation), but not before shortly dropping to about 4 km altitude in the infinite-length squall line simulations (at about 60 minutes). The strength of the upper tropospheric divergence signal gradually increases towards the end of each simulation.

Moving to the right panels with the finite-length squall lines, a substantial amount of ensemble spread is identified (earlier discussed ensemble standard deviations of divergence overlap with zero over typical depths of about 700 m, smaller than for the infinite-length squall lines). The outflow divergence has settled to levels of about 7.5 to about 13 km quite soon and remains at those levels along the outer parts (at the edge of the finite-length squall line). The convergence zone at low levels seems to slowly lift with time in this ensemble, reaching an upper bound of about 7 km after 100-120 minutes. The divergence signal seems to be much stronger in the edge region of the finite-length squall line than at its center as well. Even though the time evolution of divergence (convergence) in the finite-length squall line center simulations shares many similarities with the infinite-length squall line, the first hour has a contrasting evolution. Signals are nonetheless rather weak during that hour.

The lower boundary of the integration mask for the upper tropospheric divergence is best set at 7 km, as this boundary is most suitable differentiating the regions of convergence and divergence. Therefore, results using this altitude threshold are used for the analysis of the next Section (3.4). The upper boundary is quite stagnant at 13-14 km altitude and therefore the upper boundary is set to 14 km, which is about 2 km above the tropopause (see also Figure A1).

Note that the figures illustrate the ensemble spread. That means, they exclude simulations with physics perturbations. Especially for these perturbed simulations (notably -40% latent heating constant) the vertical profiles may differ, due to lower tops of the convective clouds. Corresponding extra panels for these simulations are available in the supplement (Figure S3).

### 3.4 Mass divergence and net latent heating ratio

Figure 5 presents the integrated strength of divergent outflows as a function of net latent heating by precipitation. Purely focusing on the three main scenarios (see leftmost three of the four green boxes in Figure 1), the separation between the infinite-length squall lines on the one hand and both the supercells and regular multicells on the other hand at given net latent

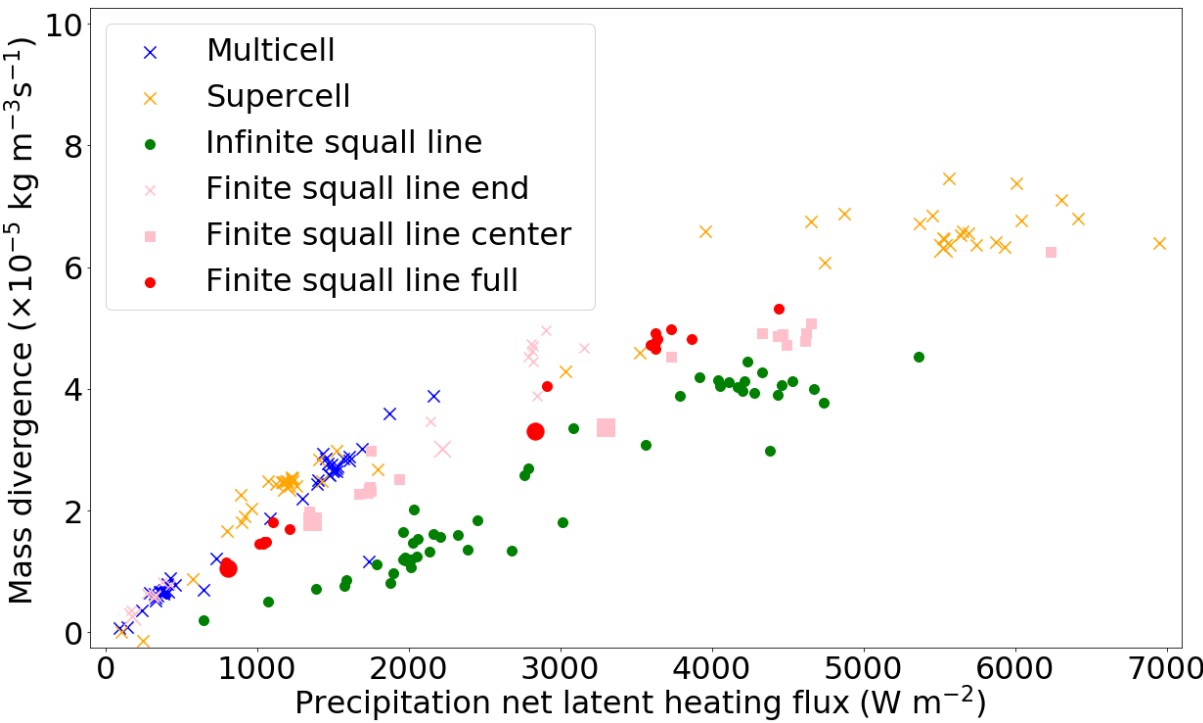

**Figure 5.** Full dataset of upper tropospheric mass divergence integrated over the layers 7-14 km versus net latent heating. Included are 206 records covering 4 modes of convection during two time intervals. The larger symbols indicate simulation data from extended domain simulations (8 in total).

heating is obvious. The latter two have increased divergence compared to all the squall line simulations. Nonetheless, this contrast is reduced at higher latent heating rates (i.e. over roughly 3000 W m$^{-2}$).

More specifically, the initial ratio between mass divergence and column latent heating is much higher for the regular multicells and supercells than for the squall lines. For increased precipitation rates, the ratio apparently decreases compared to that at

low latent heating rates. On the contrary, the same ratio for squall line even increases with precipitation intensity, although this not so clear. The robustness of the increment in the ratio, between mass divergence and precipitation rate, of squall lines is questionable. The typically lower ratios for the supercell at high precipitation rates reduce the gap between the two regimes at higher latent heating rates: on the one hand a regime that the supercell and regular multicell seem to follow and on the hand a other regime that the (infinite-length) squall line seems to follow.

Interestingly, the physics perturbations do not substantially affect the suggested regimes and resolution also has no noticeable effect in the plane of Figure 5 (it should be noted that simulations with physics perturbations and a different resolution than the control runs are included in this Figure!). The isolated convection regime (i.e. nearly 3D-outflow, applicable to regular multicells and supercells) is surprisingly linear at precipitation intensities up to about 2000 W m$^{-2}$. Moreover, the contrast in typical ratio of mass divergence and latent heating rates ("normalised mass divergence") between the two regimes that are

identified is suggested to exceed a factor two: the steep sloped line that would describe the two isolated modes of convection reaches an upper tropospheric divergence of almost $4 \times 10^{-5}$ kg m$^{-3}$s$^{-1}$ at 2000 W m$^{-2}$, whereas only one datapoint for the infinite-length squall line reaches about $2 \times 10^{-5}$ kg m$^{-3}$s$^{-1}$ at about 2000 W m$^{-2}$ ($\approx 3mm/h$) precipitation rate.

When one now focuses on the finite-length squall line simulations (pink/red, Figure 5), its end region data points (i.e. pink crosses) are in line with the mass divergence of supercell simulations. At low precipitation intensities, the mass divergence

increases as rapidly as for the supercell. In addition, both supercells and end regions of finite-length squall lines have a ratio between mass divergence and precipitation rate that reduces at higher precipitation rates in very similarly ways.

That decrease of the ratio between mass divergence and precipitation rate with higher precipitation rates also occurs for central regions of the finite-length squall line (i.e. pink squares), but the mass divergence is systematically reduced compared to the end regions of the finite-length squall lines. Hence, the behavior of the finite squall line centers aligns better with the infinite-

length squall line simulations. In spite of that, the mass divergence is systematically reduced even stronger compared to the finite squall line centers in the infinite-length squall lines.

In general, the full domain integration of the finite-length squall line leads to intermediate behaviour, with the amount of divergence in between the center and line end. That is a consequence of averaging over the center and end regions. Reduced normalised mass divergence at high precipitation intensity also occur for the full integration over the finite-length squall lines.

infinite-length squall lines represent nearly 2D convection (e.g. as in Moncrieff, 1992). On the other hand, initially isolated convective cells that are circular in an environment without convection follow the 3D-regime (supercell, multicell). At increasing precipitation or latent heating rates, outflows from deep convective cells are more likely to collide. They mimic idealised point or line sources less well. This effectively creates an intermediate dimensionality between 2D and 3D. The shift from or variability between purely 3D, intermediate, and purely 2D convection largely explain the variability in divergent outflows as

detected. It could be seen as a non-linear effect on divergent outflow from deep convection by convective organisation and aggregation. On the other hand, the absence of effects of upscaling convective organisation could have lead to a better resemblance of idealised 2D- or 3D-regimes.

The initial ratios between precipitation intensity and mass divergence suggests that the low precipitation intensity can obey to two limit regimes:

– a 2D-outflow-regime with reduced mass divergence

    – a 3D-regime with comparatively increased mass divergence in the outflow region

The quasi-2D regime occurs as neighbouring cells compensate each other's divergent outflows efficiently in squall lines. Even though three dimensional outflow from the individual cells in a squall line exists, the meridional component is largely compensated by the neighbouring convective cells, as they also produce (opposing) meridional outflow. In the summation over the

divergence of all convective cells, these meridional components compensate and produce no net divergence along the y-axis of the simulations. Therefore, the zonal component of the net divergence is an order of magnitude larger in an infinite-length squall line than the meridional component (next Section, Section 3.5). The ratio between mass divergence in the two regimes could be around a factor of three according to the dataset of Figure 5! At higher precipitation intensities these regimes are

not obeyed in our simulations, as mentioned. This means that a line at intermediate normalised mass divergence between the two regimes might exist, around which all data points scatter at high precipitation intensities. In the finite-length squall line case, the mixed or intermediate dimensionality is already effective at low precipitation intensity. This concept and qualitative explanation connects the findings of the LES-simulations with Bretherton and Smolarkiewicz (1989), Nicholls et al. (1991), and Mapes (1993).

Investigating the set of larger symbols in the scatter plot (Figure 5) - those symbols that represent a supercell simulation at an extended domain - they appear within the range of data points covering the perturbations. Similarly, for the finite-length squall line simulations the mass divergence to net latent heating ratio is often slightly lower than that associated with the ensemble mean. Nevertheless, divergence in extended domain simulations is never substantially outside of the range within each mode of convection.

The ordering of the different modes of convection - with regular multicell, supercell, and finite squall line ends inducing most mass divergence at a given precipitation rate, followed by the full finite-length squall line, the finite-length squall line center, and, lastly, the infinite-length squall line with weakest mass divergence - is hardly affected by any of the perturbations among all simulations (see Figure 1 for an overview of the perturbations). That ordering indicates that mass divergence at given precipitation rate depends on the organisation of a convective system. Initially, it is suggested to manifest as a 2D-outflow-regime (line source) with weaker mass divergence on the one hand and a convective 3D-regime (point source) enhances mass divergence on the other hand. At high precipitation rates and over the course of time, the convective systems do not stick to these idealised regimes.

By focusing on the multicell divergences in Figure 5, it can be induced that contrasts in the divergence relative to the precipitation between the first and second time interval do not or hardly exist. Based on this argument and closer inspection of the spatial distribution of divergence patterns in the simulation dataset, it is found that the precipitation (heating) pattern is solely responsible for the region of upper tropospheric divergent flow.

## 3.5 Zonal and meridional divergence components in finite squall line

The existence of two outflow regimes (2D and 3D divergence) has been

- suggested in the previous section

- suggested by analytical expressions of flow perturbations derived from a linearised gravity wave model forced by heating (Bretherton and Smolarkiewicz, 1989; Nicholls et al., 1991; Mapes, 1993; Pandya and Durran, 1996; Nascimento and Droegemeier, 2006)

- documented for related pressure perturbations and updraught strengths by Morrison (2016a, b)

In this section the $u$- and $v$-component of the upper tropospheric divergence are separately investigated for the finite-length squall line.

Figure 6a, b shows that during the initial time interval, the meridional component of the divergence is negligible throughout

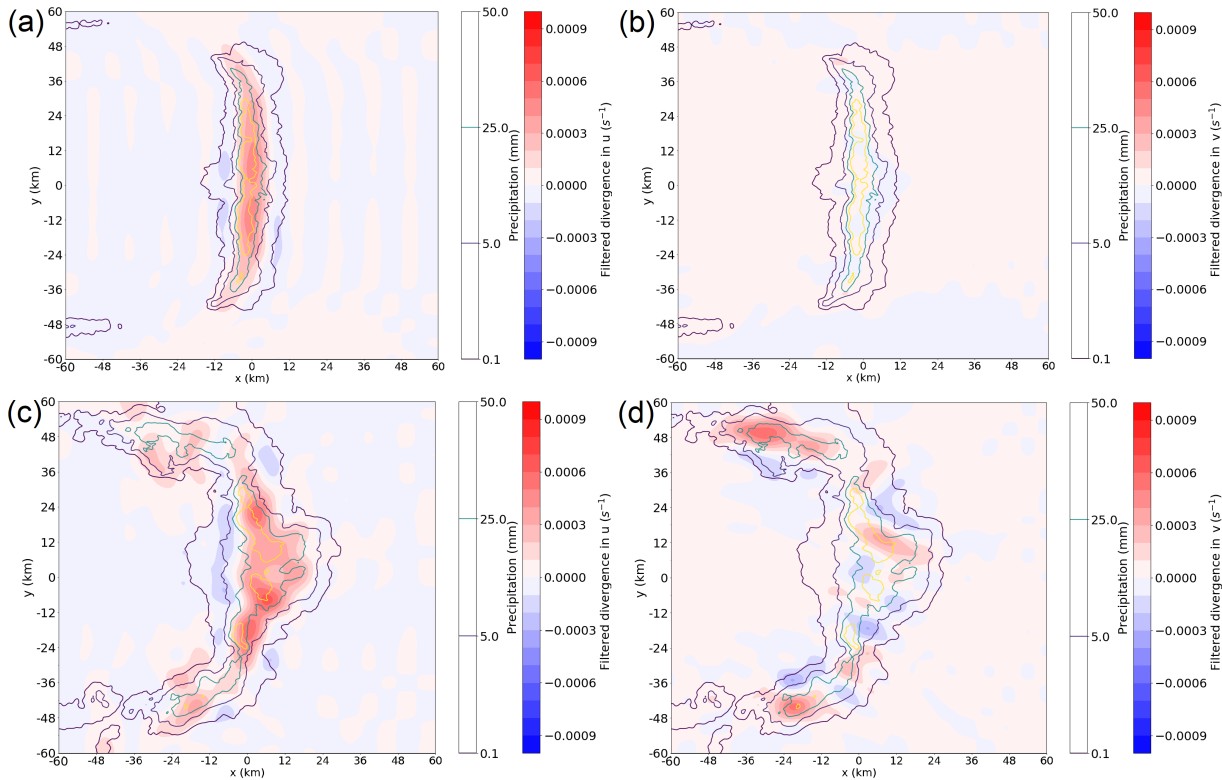

**Figure 6.** Spatial distribution of filtered divergence over altitudes 7-14 km for a finite squall line during (a, b) first and (c, d) second time interval. Wavelengths that fit more than 20 times in the domain have been removed with a discrete cosine transform. Contours indicate the accumulated precipitation pattern during each of the two time intervals (0.1, 5, 25, 50 mm, as displayed in one of the two colour bars). Both the zonal (a, c) and the meridional divergence component (b, d) are displayed separately.

most of the domain. The pattern in the $v$-term of the divergence is consistent with this part of the finite-length squall line resembling the 3D-outflow-regime. Similar patterns have been identified for the infinite-length squall line (not shown), but without an enhanced meridional component of divergence at the squall line end there. Practically, all divergence occurs initially in the $u$-component, which is consistent with the manifestation of a 2D-like outflow regime in the infinite-length squall line

and the centre region of the finite-length squall line (Figure 5).

During the second time interval, the $v$-components develop particularly strongly on the northern and southern arching regions of the squall line. Both components are of the same order of magnitude, but that is to be expected for a strongly curved squall line. The pattern of squall line curvature is clear from the accumulated precipitation (and, similarly, the simulated radar imagery in Figure 2l). The possibility that outflow collisions occur and the effective flow resembles a regime governed by a mixture of

410 2D- and 3D-regimes - as a result of interference between outflows from individual convective cells - is strongly supported by both Figure 5 and 6. Hence, Figure 6 provides further evidence for the explanation of Section 3.4.

The (mean) divergence in Figure 6a is about an order of magnitude larger than in Figure 6b in the squall line centre. This is also

consistent with the infinite-length squall line, where the difference is about one order of magnitude too. In the bottom panels of the figure, that contrast has strongly reduced as a result of the arching squall line - accordingly with expectations of bending mixed 2D-3D convection. Therefore, the leading order divergence variability in Figure 5 can confidently be attributed to the dimensionality of outflows (2D/3D), and, correspondingly, the presence of a pulse or line source of latent heating.

## 4   Discussion

### 4.1   Detection of outflow: spatial and temporal analysis windows

Upper tropospheric outflow from deep convection has been quantified for each simulation by integrating the mean mass divergence in 3D over a region surrounding the convective cells (horizontally) and over 7-14 km altitude. Figure 4 and divergence profiles (not shown) revealed that the lower boundary is the most critical of the two. Therefore, the integration has been repeated for the 6-14 km layer. The resulting patterns in a latent heating versus divergence diagram comparable to Figure 5 are not sensitive to the lower boundary.

The procedure has been executed for two time intervals separately: a first time interval where the fastest gravity waves escape the box of integration and a second where a large proportion of the gravity wave signal has escaped that box. Even though some potentially relevant flow effects with consequences for upper tropospheric (UT) divergence could have escaped this integration box with the gravity waves, the results suggest that this is not the case. That such an escape did not have consequences for diagnosed UT divergence can be justified with the following arguments:

– The mechanism of gravity waves is to restore density anomalies with fluctuations, which are averaged out when integrated over longer distances in the quasi-horizontal plane.

– The similarity of the ratio between mass divergence and net latent heating (1) during the first and second time interval for regular multicells and (2) the similarity of that ratio to supercell simulations during the first time interval suggests that an escape of gravity waves plays no role to the mass divergence. It certainly does not dilute the outflow divergence by underdetection.

– Mass divergence of updraught outflow cannot initiate at a location outside of the convective cell's updraught itself and spatial distributions of the winds and cumulative precipitation (e.g. Section 3.5) support that argument.

– Related to the previous argument: testing of different integration masks for the large domain supercell simulation indicates that mass divergence only decreases relative to the precipitation rate (!) when integrating over a too large domain. This is a sign of substantial increase in subsidence within the integration mask as soon as the mask is altered; the subsidence develops with gravity wave propagation (see Bretherton and Smolarkiewicz, 1989; Mapes, 1993, and the supplementary material). The tests essentially suggest that the essential divergent outflows are included by including the precipitation cores within the integration mask. That is consistent with the spatial distribution of our divergence signals

and with the linear gravity wave adjustment model triggered by convective heating patters, as documented in Bretherton and Smolarkiewicz (1989) and Nicholls et al. (1991).

An integration mask covering the convective cores and ending just outside of the area of precipitation accumulation leads to the detection of a large proportion of the divergent outflows. Little dilution from convergence/inflow may occur if appropriate vertical levels are selected for vertical integration.

## 4.2 Deviations of perturbed simulations from main UT divergence structure

### 4.2.1 Physics perturbations

In Figure 5 one can see a very robust signal of convective organisation with significance for the divergent ouflow. However, a few odd datapoints occur. By design and nature, the strongest physical perturbations (e.g. -40% latent heating constant) suppress the deep convection and while some precipitation occurs in these perturbed simulations, the divergent outflow does vertically not fit in the integration mask as well as for the ensemble and the large majority of other simulations. It is verified that data points appearing as outliers in Figure 5 shift toward those of similar organisation type, if the vertical mask of divergent
outflow integration is shifted appropriately to other vertical levels (with appropriate density weighting). An extension of Figure 4 can be found in Figure S3 of the supplement. The dataset visualised in this figure shows how the integration masks of outliers have to be shifted for better alignment of particular simulations with the general pattern in Figure 5.

### 4.2.2 Specific role of convective momentum transport

The strong order in Figure 5 directly suggests that convective momentum transport does not have a direct systematic impact on
divergent outflow from deep convection. That does not imply that convective momentum transport does not play a role at all: it can modify the convective evolution and subsequent organisation indirectly and therefore affect the precipitation intensity. The latter two do affect the divergent outflow, as the dataset presented in Figure 5 revealed. Even if some scatter in the mass divergence occurs for a given precipitation rate and convective organisation type, it does not systematically relate to increases or decreases in convective momentum transport. Such scatter mainly occurs at high precipitation rates for the infinite-length
squall line and the supercell in Figure 5.

### 4.2.3 Adjusted low level stratification

The signal of convective organisation in Figure 5 is robust. This is because the ordering of the different modes of convection is robustly present among a dataset with the background ensemble, physically perturbed simulations (in two ways), and the simulations on extended domains. Additionally, initial condition potential temperature profiles have been perturbed in an
470 additional set of simulations (see Section 2.3.3 and Figure 1) to test whether the stratification of the low levels has a substantial impact. The dataset obtained suggests that this is not the case. Similar perturbations have been used for the wide ensemble of finite-length squall lines. The structure in Figure 5 is well established and this implies that a wider range of initial conditions than just those of a specific thermodynamic profile is explored with the dataset. The data strongly suggest that magnitude of

divergent outflow relates to net latent heating in a similar way, irrespective of the strength of near-surface inversions and the magnitude of moist instability.

### 4.3 Two mass divergence regimes at low precipitation rates

Figure 5 suggests that there is about 1500 W m$^{-2}$ of precipitation equivalent needed for $1 \times 10^{-5}$ kg m$^{-3}$s$^{-1}$ of divergence at low precipitation rates in an infinite-length squall line, whereas this is only about 500 W m$^{-2}$ for the regular multicell and supercell regime. The proportionality between these two regimes is well over a factor two, and likely very close to 3 and $\pi$. The idea of the finite-length squall line simulations is that the outer part at both of the squall line ends mimics a regime where convective cells can freely induce their outflow in a 3D space when both ends are combined (as if the center is "removed"), as in the case of a multicell and supercell. The center, however, is geometrically somewhat restricted, as in that of an infinite-length squall line. Infinite-length squall lines only allow for outflow in one horizontal direction. The idea is supported by the magnitudes of two components of the mass divergence in finite and infinite-length squall line simulations: outflow in the zonal direction exceeds that in the meridional direction by an order of magnitude, consistently with findings pointed out by Nascimento and Droegemeier (2006). Similarly, in the finite-length squall line, divergent outflow is much larger in the direction normal to the finite-length squall line than in that parallel to the squall line initially, with dynamics induced by small cells (short wavelengths) compensating each other in the parallel direction (Figure 6). That provides further support for the idea of two outflow regimes.

Outflow simulations with expressions based on the numerical model of Bretherton and Smolarkiewicz (1989) and Nicholls et al. (1991) suggest that the ratio of convective outflows between a line source and point source is a factor of $2\pi$ in the limit case. That stems (in their calculations) from a conversion of the delta function from radial geometry to an x-y plane in their derivation (Nicholls et al., 1991). A mechanism that could explain the deviation of about factor two between their linear gravity wave models and large eddy cloud simulations as performed here has not been found yet. Theoretical support for different regimes of updraught and pressure perturbations between 2D (line source) and 3D (a point source) in a weak shear environment is also provided by Morrison (2016a). The updraughts and pressure perturbations as studied by Morrison (2016a) drive the outflows as studied here; outflows relate to updraughts through continuity. Morrison (2016a) derived a deviation factor of two theoretically and subsequently compared the findings to cloud simulations (Morrison, 2016b).

The robustness of the results, together with the arguments above, give high confidence in the impact of outflow dimensionality on the magnitude of the divergent winds. Furthermore, the intermediate mass divergence to precipitation rate ratio at high precipitation intensities compared to the initial 2D (low ratio) and 3D (high ratio) regimes suggest that convective aggregation likely affects the dimensionality of convective outflow in the upper troposphere. Outflow likely adapts to a mixture of 2D- and 3D-regimes due to the convective organisation and interference between outflows of individual cells. When aggregates of convective cells collide with upper troposhperic outflows of other convective cells, the effective dimensionality would be something intermediate between 2D and 3D: the outflows first collide along the line through the updraughts, and become nearly-2D along the line, but on the outer regions the outflow can still move as if the convective cell was isolated. That

corresponds to a nearly-3D outflow regime, and any mixture creates ovals of outflow similar to the finite squall line in our conceptual framework (even if the supercells also reveal such behavior and collisions of outflow after some time).

## 4.4 Implications and future research

The mass divergence found in the dataset (Figure 5) is in terms of magnitude in good agreement with linear gravity wave adjustment models where heating is imposed as proxy for a convective system (Bretherton and Smolarkiewicz, 1989; Nicholls et al., 1991; Mapes, 1993; Pandya et al., 1993; Pandya and Durran, 1996).

The initial 2D- /3D- regime behavior with reduced divergence at higher precipitation intensities due to convective aggregation found in this study contrasts with the modelling and observation studies by Mapes (1993) and Mapes and Houze (1995). In
Mapes (1993) it is suggested that a stratiform contribution by the vertical half wavelength due to a stratiform fraction of a mesoscale convection system actually increases the divergence at any given heating rate or at a given precipitation intensity. On the one hand this could imply that the convective systems simulated in this study do not extend sufficiently and generate any sizeable stratiform precipitation system. Indeed, the stratiform contribution to the simulated squall line and supercell clouds is not substantial, especially when looking at the precipitation that reaches the surface (turned into net latent heating); see Figure
2 and Figure S1 in the supplementary material (see also Groot and Tost, 2023). The formation of a stratiform precipitation regime usually coincides with lifting of the level of neutral divergence, on average (Mapes, 1993; Houze, 2004). In Figure 4, such a continuous gradual rising of the level of neutral divergence for supercells and squall lines during the second hour cannot be detected. However, at least a fraction of an MCS is usually stratiform within the second hour. This suggests that the simulations are only representative for purely convective systems or those with minor stratiform fractions, which is a reason-
able approximation to convection in certain regions (Schumacher et al., 2004). In spite of this, these purely convective systems are important to study for a better understanding of the role of deep convection in the climate system and to improve deep convective parameterisations.

On the contrary, using Figure 5, one could argue that a stronger increase of the mass divergence with latent heating rate occurs for the infinite-length squall line simulations at rather high latent heating rates than at low latent heating rates (as the
530 highest latent heating rates occur in the second hour). Furthermore, stratiform precipitation system may form out of the squall line anvils. That would be in agreement with arguments by Mapes (1993) and Mapes and Houze (1995), assuming that the stratiform fraction of the system should increase with time. However, an infinite-length squall line is in practice not very representative for most deep convection in the real world. In addition, the level of neutral divergence does not seem to rise at all for the infinite-length squall line in Figure 4, which would not support arguments of Mapes (1993) and Mapes and Houze (1995).
Furthermore, the more realistic finite-length squall line is not behaving in agreement with these studies. In the finite-length squall lines, the ratio between mass divergence and net latent heating decreases at higher precipitation rates during the second (later) time interval (Figure 5).

Theoretical 2D squall line models have been extensively studied by Moncrieff and co-authors (e.g. Moncrieff, 1992). In Moncrieff (1992) it was argued that such 2D models could be very beneficial for parameterising convective momentum transport.
Trier et al. (1997) pointed out that one should be careful and that processes such as convective momentum transport in actual

squall line convection can have characteristics of a mixture of 2D and 3D convection. Certain sections have more characteristics of a 2D convection regime and others more of a 3D convection regime. The finite squall lines suggest that the same is true regarding divergence profiles. That means that a comparison of traditional and theoretical models of idealised convection (e.g. 2D models) to cloud resolving and large eddy simulations is needed. By stimulating such model intercomparisons, the applicability of traditional practices and findings to the more complex simulation techniques can be scrutinised (as done here and in Morrison, 2016a).

Theoretically, convectively induced divergence profiles are suggested to mimic 2D- or 3D-regimes in some cases. On the contrary, in practice intermediate behavior is suggested to be more likely, especially for intensive systems. That can be a worthwhile consideration for the development of convective parameterisations that would take convective organisation and aggregation into account (e.g. Moncrieff, 2019).

In Baumgart et al. (2019) and Zhang et al. (2007) it was found that numerical weather prediction errors are initially established predominantly in regions of enhanced and mostly convective precipitation. Baumgart et al. (2019) were able to attribute initial error growth ($< 12$h into the simulation) in their stochastically perturbed simulations to non-conservative processes and predominantly to the deep convection parameterization. That parameterization represents the collective effect of organized convective systems and isolated convective cells. At later times, the induced ensemble variability corresponds predominantly to variability in upper tropospheric divergent winds. Baumgart et al. (2019) inferred that this upper tropospheric variability is likely associated with latent heat release below and corresponding deep convection as precursors.

Quantitative understanding of upper tropospheric outflow and uncertainty quantification achieved in this work could support an extension of the potential vorticity diagnostics of Baumgart et al. (2019) towards smaller scales. Consequently, it may lead to better insights in the role of individual convective systems in certain forecast errors. Furthermore, it may reveal biases between certain modelling approaches, namely large eddy simulation, cloud resolving simulations with explicit deep convection, and global simulations with parameterized deep convection (see for example Done et al., 2006). Intercomparison of divergent outflows among the simulations investigated here and various simulation set-ups could provide structural insights in (potential) biases between each of them. Such insight in structural biases depending on the treatment of deep convection may be beneficial to the understanding of weather and climate simulations. In this work, a set-up for intercomparison of divergent outflows is established, focused on large eddy simulations with advanced small-scale process representation. Effects of differential convective organisation or differential diabatic heating could potentially be followed to synoptic scale uncertainty days ahead, using thorough understanding of upper tropospheric divergence variability (e.g. Baumgart et al., 2019; Rodwell et al., 2013). This study builds the insights into which main factors control the upper tropospheric divergence, which is required to trace diabatic heating mesoscale uncertainty cascading upscale to synoptic scale uncertainty.

## 5  Conclusion

LES simulations of four types of convection with ensemble, physics, resolution and other modifications have shown that upper tropospheric mass divergence associated with deep convective systems depends on net latent heating (i.e. precipitation

intensity) and on the convective organisation. Divergent outflows have been integrated over a fixed area and 7-14 km altitude.
Thereby, the main precipitation cores for each type of convection have been covered and split over two time intervals (Figure 3). Wind profiles were imposed such that the convective cells propagate slowly with respect to the domain, and therefore their flow perturbation accumulated in a condensed region. The relation between mean mass divergence and net latent heating rates has been analysed and intensively discussed:

At low precipitation rates and in initial development stages, the four basic scenarios strongly suggest the existence of a 3D-outflow-regime for isolated convective cells (i.e. supercell and regular multicell). On the other hand, a 2D-outflow-regime exists, as demonstrated by the infinite-length squall line. Within each regime, a linear dependence of mass divergence on net latent heating is found. In practice, a more realistic squall line is suggested to conform to a mixture of the 2D and 3D-regimes. At higher precipitation intensities, the outflow does not strictly follow these two regimes anymore. The magnitude of the mass divergence associated with high net latent heating rates is somewhat intermediate between the initial 2D- and 3D-regimes, but ordering between different organisational types of convection still occurs at high net latent heating rates. Given an outflow dimensionality, UT divergent convective outflow therefore depends linearly on net latent heating accordingly. This dependence is the key finding of this study. Additionally, non-linear dependence occurs through convective aggregation and organisation that can modulate outflow dimensionality.

Simulations on extended domains strengthen the confidence in the results, in addition to an ensemble of simulations. Convective momentum transport plays no direct role for the mass divergence in the simulations. Nevertheless, by affecting the organisation and precipitation rates within a convective system, it can indirectly affect upper tropospheric mass divergence.

An important implication of this study is a potential bias in upper tropospheric divergent flow if the convective organisation is unknown or misrepresented (as it is generally the case for parameterisation). This implication exists even if the precipitation rate or some kind of statistical distribution describing the spatial mean and variability of precipitation rate would be known.

The findings are in good agreement with the linear gravity wave adjustment models triggered by convective heating in Bretherton and Smolarkiewicz (1989), Nicholls et al. (1991), Mapes (1993), and with Morrison (2016a, b), if we assume a purely convective heating regime. However, the simulations of this study do not reveal any substantial heating contributions from stratiform fractions of mesoscale convection (Mapes, 1993). Therefore, anything specific regarding the role of vertical modes other than that of the basic convective heating profile triggering wavelength $2H$ (twice the depth of the troposphere) cannot be concluded. Additional simulations beyond the scope of this study would be necessary to understand the role of secondary gravity wave modes affecting UT divergence. Simulations tackling the effect of those modes would have to be focused on the finite and infinite-length squall lines at larger domains specifically. Longer integrations are needed to inspect the effect of those modes.

*Code availability.* Part of the output of this study are available for download under Groot (2022) (accompanied dataset of Groot and Tost (2023); last accessed: 10-01-2023) and other parts under Groot (2023) (last accessed: 10-02-2023). A download script and README is provided there.

*Author contributions.* The idea for this study originates from the authors in collaboration with colleagues from TRR 165. The study was designed, conducted and composed by EG, with contributions from and under the supervision of HT.

*Competing interests.* HT is also a co-editor of this journal. However, this does not represent a competing interest for this publication; there are no further competing interests.

*Acknowledgements.* The research leading to these results has been done within the subproject 'A1 - Multiscale analysis of the evolution of forecast uncertainty' of the Transregional Collaborative Research Center SFB / TRR 165 'Waves to Weather' funded by the German Research Foundation (DFG). HT acknowledges additional funding from the Carl-Zeiss foundation. The authors would like to thank the collaborators in Waves to Weather project A1, namely George Craig, Michael Riemer and Tobias Selz, for their input and, in addition, those who helped the lead author to facilitate the initialisation of CM1 on the high performance computer in Mainz: Manuel Baumgartner (now at DWD) and Christopher Polster. Lastly, we would like to thank the reviewers for their useful suggestions and the editor for handling the manuscript. The authors would also like to acknowledge the computing time granted on the supercomputer MOGON 2 at Johannes Gutenberg-University Mainz (hpc.uni-mainz.de, last accessed: 10-01-2023).

## Appendix A:  Initial condition profiles

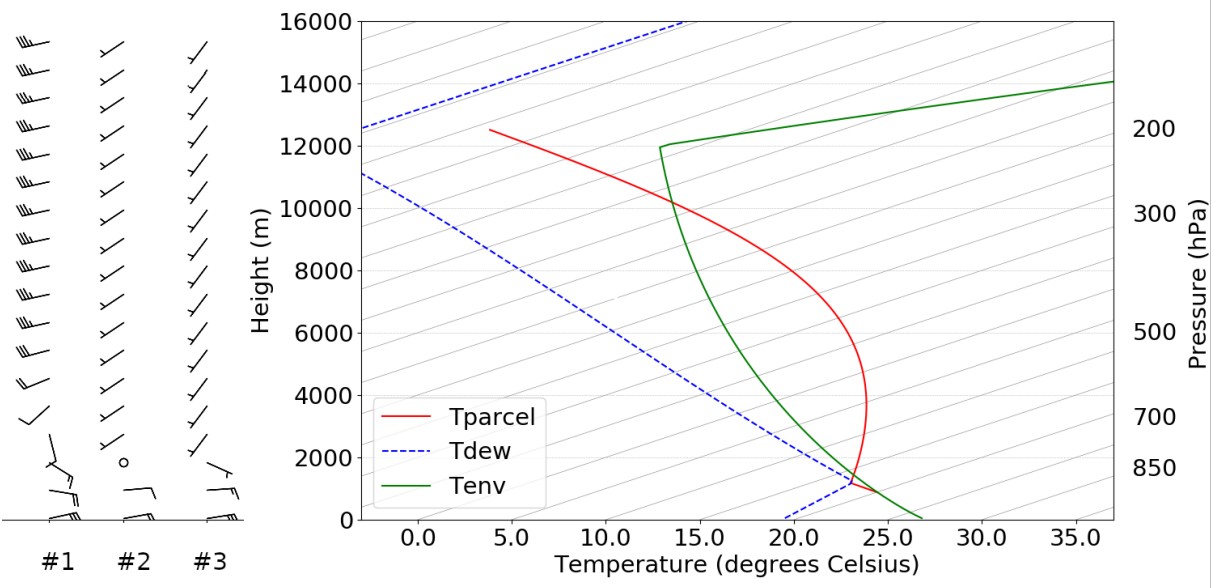

**Figure A1.** Temperature and moisture profile following Weisman and Klemp (1982) and wind profiles 1-3 (left). Temperature: green solid line; dew point: blue dashed line; temperature of parcels when lifted from about 900 m altitude (no dilution): red line.

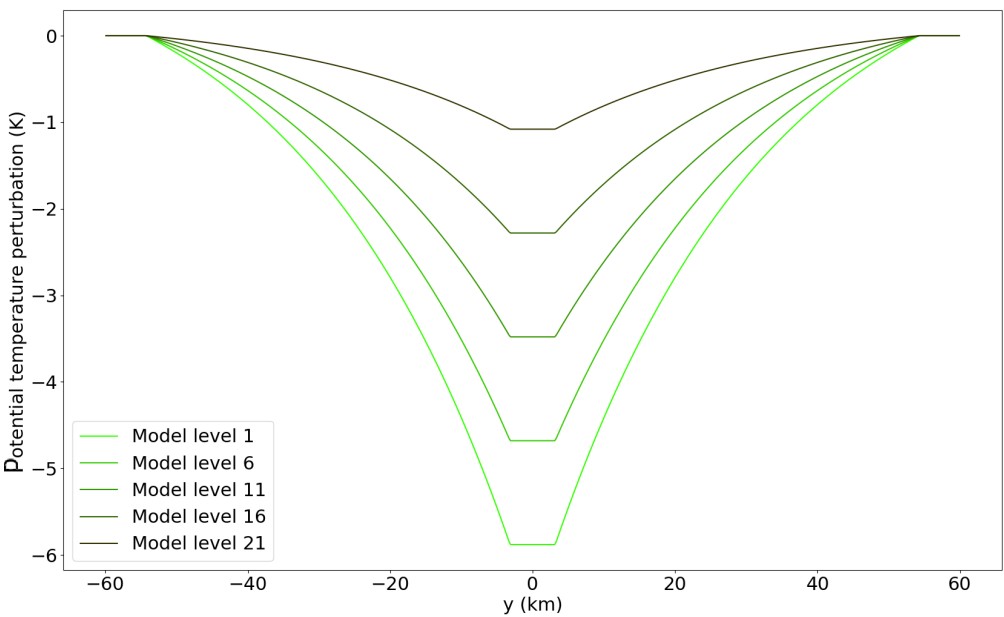

**Figure A2.** North-south profile of initial potential temperature perturbations along the length of the finite-length squall line for five selected model levels, counted upward from the surface level.

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
