# Peer review of "Divergent convective outflow in large eddy simulations"

_EGUsphere, 2022_

## Referee Comment (RC1)

General comments:

The entire paper is well written, cites relevant references, and has in my opinion a very nice balance between presenting strong results and being self-critical about them. The introduction gives a clear motivation for the work and realistic expectations on what the reader can expect. The remainder of the paper is clearly structured and easy to follow. There is an extensive discussion, which is highly interesting. The implications for modelling on any scale (aside from our theoretical understanding of convection), from LES to global climate are substantial. This work connects (rather difficult) theory with practical implications and should be of great interest to the weather and climate community. I can recommend to publish this work and only have a few ideas for improving it further, which I list below.

Specific comments:

1) The model setup misses a description of how boundaries are treated
2) The caption of Fig. 1 mentions "this Section (Section 2.3)", but the figure is first referred to in Section 2.2. At this point I was not able to understand the figure. In the end I still struggle. Could this figure be improved so that it better explains the different ensembles and the logic? That could be achieved perhaps by naming the relevant sections in the figure. Or by a table, which could have a column with comments like: "to test this and that"…, "same as experiment XY but with modified XZ"
3) L383 and 400: how can we distinguish the first and second time interval? I understood that Fig. 5 uses the same symbols for both. I was wondering if that is bad. Can the authors comment on it?
4) The other comment that I was waiting for is the apparent leveling off at constant divergence in Fig. 5 at the highest heating rates. What is the interpretation?
5) L404: "Mass divergence of the outflow cannot originate from a point outside of the convective cell's updraft itself": I don't understand what the authors mean by this. What physical process does this reflect and what other possibility would there be? Also: the compensation of neighboring cells was discussed. Is this sentence not a contradiction?

Technical comments – or very minor

1) Fig. 3: what does white color mean?
2) L274-275: I do not understand this sentence: "The box size over which outflow is diagnosed has to be sufficiently as a result of that shortwave pattern." Sufficiently what?
3) L278: what could cause the boundary effects?
4) Generally: the units are not correctly formatted, neither in the text nor in figures.
5) L87: reference format
6) L89: suggest "occurs" -> exists

---

## Referee Comment (RC2)

**Summary:**

This study analyzes the relationship between upper – level divergence and latent heat/precipitation using numerous LES simulations of supercells, multicellular convection, or squall lines. It is found that the amount of upper – level divergence is primarily dependent on the amount of latent heat/precipitation and storm morphology. It was also found storm morphology determined whether the outflow was two or three dimensional. These results are consistent with linear wave theory. The authors point out that their model did not adequately simulate stratiform precipitation, so their results only apply to convection with small anvils and say nothing about how other heating modes would impact the results. I found this manuscript very difficult to read and follow the logic used in the manuscript. The analysis was unclear or lacking details and discussions were too overly broad. These points are highlighted in more detail below.

**Specific Comments:**

1. The manuscript has 5 figures and at least 5 supplemental figures and numerous references to material in the supplemental material. This was done to the extent that it seemed that an entire portion of the analysis is done in the supplemental. For example, section 4.2 discusses the impact of physics perturbations and almost entirely references Figure 5. However, Figure 5 only shows data by storm type. There is no indication of how the relationship changes with momentum or low – level stratification. Thus, I don't think Figure 5 can be cited as proof of the arguments in this section. I suggest either figuring out how to add this information to the figures included in the manuscript or break this manuscript into two separate manuscripts.

2. Line 71: It is mentioned that the manuscript the diabatic forcing and resulting flow will be tracked using the methodology from Baumgart et al. (2019). What exactly is that methodology? Baumgart is only referred to the Introduction and nowhere else in the manuscript.

3. How are model boundaries treated in the model? Are some of the wave patterns seen in the vertical velocity due to reflection off the edges of the model? I assume this is described in Groot and Tost (2022), but it is a very important detail related to these runs and should be mentioned here as well.

4. The abstract says nearly 100 ensemble members are created. The manuscript indicates several ways the simulations were perturbed, but I don't see how the ensemble and physics perturbations listed resulted in 100 ensemble members.

5. Are the regions over which the diagnostics integrated over the same area/size? If they are different, how could averaging over different areas impact your results? Could some of the regime differences be attributed to the area difference.

6. Figure 4 and its analysis is very confusing. When I looked at the figure several statements that were made in the manuscript did not seem consistent with the figure. For example: not everything developed intense convection within 30 mins (line 286) and not everything in the top and bottom row has strong convergence within 45 minutes (line 290). I also don't see the white line dipping down to 4 km (line 304). In general, I suggest referring to the type of storm in each panel not its row or column position. Please explain the white lines earlier. Additionally, is the data above 14 km really needed?

7. Section 3.4: Talks a lot about changes in slope. It is very hard to visualize. Please create the regressions, show the values of the slope, and prove that they are significantly different using $R^2$.

8. One of the main conclusions is that there whether the outflow is two dimensional or three dimensional is determined by storm morphology. There was no figure that explicitly showed it. If something is included in the abstract and conclusions sections, it needs to be proven with a figure in the manuscript – not in the supplemental.

**Technical Comments:**

1. I found the sentence structure in this manuscript to be convoluted. The sentences are very long and use numerous commas. While present throughout the manuscript, the introduction was particularly difficult to read. I encourage the authors to break their sentences into small, more succinct sentences. Here are just a few examples:
   a. Lines 46 – 51, Line 262 – 263,
2. I was a bit confused by the organization of the introduction. Lines 1 – 44 and 57 – 74 seem to be background information. However, lines 45 – 56 state the objectives of the paper. This organization felt jumpy and caused me some confusion. I suggestion combining lines 1 – 44 and 57 – 74 and then stating the objectives and outline of the paper.
3. I suggest using the oxford comma throughout the manuscript. There are numerous commas throughout the paper. However, without the use of the oxford comma I found myself getting confused as whether something was a list or an aside.
4. Sometimes it is suggested that 3 convective regimes are evaluated, but sometimes it is said that 4 convective regimes are evaluated. Please be consistent.
5. Lines 24 – 26: I disagree with this statement. It has been shown that, overtime, stratiform regions can create just as much or more latent heat release than convective regions. Stratiform regions have lower precipitation rate than convective regions. This is especially true for mesoscale scale convective systems, which often include squall lines. Also, what does "(intuitively)" mean, imply here?
6. Line 30: What is the name or type of the model used?
7. Line 53: Which three factors? Latent heat, momentum, and organization or the control, ensemble, or physics perturbations?
8. Line 70: What impacts is being referred to here?
9. Lines 144 – 148: Does this apply to all storm types or just a subset?
10. Line 186: Why was the simulated extended to 160 minutes when data was analyzed only out to 120 hours?
11. Line 194: This sentence seems out of place. It talks about area section, but the rest of the paragraph talks about time selection.
12. Please add labels or letters ("a") to the panels in figures for easier referencing.
13. When referring to higher latent heating rates please be specific. What I deem high maybe different than you.
14. Line 333: What is the isolated convective regime? It has not been defined yet.
15. Line 353: remove the double "is"

---

## Author Comment (AC1)

**Replies to all Referee Comments on "Evolution of squall line variability and error growth in an ensemble of LES " [1]**

Edward Groot, Holger Tost

March 6, 2023

**1  General**

We thank referees 1 and 2 for filtering out and formulating their pointwise summary and feedback [2, 3]. The manuscript plus its supplement have been restructured and mostly linguistically improved according to the referees' comments.

**2  Referee 1: reply to RC1 [2]**

*'The model setup misses a description of how boundaries are treated'*

The boundary treatment has been clarified in the model set-up Section (2.1) and also follows [4]. See also [3] and the reply to their comments.

*'The caption of Fig. 1 mentions "this Section (Section 2.3)", but the figure is first referred to in Section 2.2. At this point I was not able to understand the figure. In the end I still struggle. Could this figure be improved so that it better explains the different ensembles and the logic? That could be achieved perhaps by naming the relevant sections in the figure. Or by a table, which could have a column with comments like: "to test this and that"..., "same as experiment XY but with modified XZ"'*

The Figure caption has been clarified. From the caption (updated manuscript [1]):
"Overview of all CM1 experiments done in this study. The four scenarios represented in the columns of the display have been introduced in Section 2.2. These green boxes with white caption show each of the four prototypes of convection that we use, with a list of experiment groups following in the column below each of them. Below the column header, the perturbations applied to each scenario are represented, which are discussed in the order of display, downward (Section 2.3). Here, white text represents the regular basis set of perturbation experiments applied to the first three modes of convection and black fonts represent irregularities among the experiments, tailored at specific modes of convection and robustness testing".
To complement the update, some additional context follows below (consider this as something similar to a footnote):
The white font experiments were needed as basis.
The experiments in black are not applied regularly in the same way to all of the first three scenarios, since as a result of irregularities in initial conditions or convective evolution, choices have to be altered slightly between the three modes of convection. In case of the low-level stratification, experiment regularity is for example not possible, due to differences in the initial $\theta$-profile between squall lines and supercells/regular multicells on the other hand. Furthermore, in case of the extended domain simulations, they are not useful, since only supercells and finite-length squall lines are curved systems that are affected differently (by boundaries, domain size) along the system orientation: as a result of their curvature.
As infinite-length squall lines are symmetric along the $x = 0$-line, the effect of the domain size is not interesting, even though they also evolve close to the boundaries. Furthermore, ordinary multicells are very small and boundary effects or domain size are not of any concern.

*'L383 and 400: how can we distinguish the first and second time interval? I understood that Fig. 5 uses the same symbols for both. I was wondering if that is bad. Can the authors comment on it?'*

Yes, this is true. The choice was made to not differentiate between the two intervals, as the main message of the figure and paper is that storm type/outflow dimensionality causes the differentiation in mass

divergence. The comparison in Figure 5 consists of many different classes and only the storm type has been chosen to be distinguishable to emphasize this message. No difference occurs for the multicells in the first interval and the second interval and no structural differences are found other than the main ones identified in the manuscript between the first and second interval (as mentioned in the discussion).

By making the underlying data of Figure 5 available in the updated supplement, it is possible for very interested readers to analyse this difference themselves, if they want.

*'The other comment that I was waiting for is the apparent leveling off at constant divergence in Fig. 5 at the highest heating rates. What is the interpretation?'*

This is meant with the following:
"When aggregates of convective cells collide with upper troposhperic outflows of other convective cells, the effective dimensionality would be something intermediate between 2D and 3D: the outflows first collide along the line through the updrafts and become nearly 2D along the line, but on the outer regions the outflow can still move as if the convective cell was isolated. That corresponds with a nearly 3D outflow regime, and any mixture creates ovals of outflow similar to the finite squall line in our conceptual framework (even if the supercells also reveal such behavior and collisions of outflow after some time)" (lines 475-480)[1].

In other words, the effective/apparent dimensionality of the convective system seems to decrease as various outflows collide and push against each other, leaving the initial 3D regime. It could be seen as an effective dimensionality going from 3.0 to for example 2.6, 2.5 or 2.4, with increasing precipitation intensity. As a result of more cells needed to produce higher precipitation rates over a large area, the effective divergence per unit latent heating in the column will decrease.

*'"Mass divergence of the outflow cannot originate from a point outside of the convective cell's updraft itself": I don't understand what the authors mean by this. What physical process does this reflect and what other possibility would there be? Also: the compensation of neighboring cells was discussed. Is this sentence not a contradiction?'*

The authors are not exactly sure about the precise interpretation of this question. Nevertheless, we have slightly reformulated the sentence, which hopefully clarifies it.

The idea of the argument presented in the main manuscript is that, following the understanding that [5, 6] provide (and follow-up studies cited in the main paper), gravity wave dynamics initiates a storm-relative flow at upper levels: divergent outflow. This gravity wave starts at the location of the updraft, which is, under the presence of not overly strong storm-relative winds, right above the region where precipitation accumulates. The divergent outflow can escape as a gravity wave, but as long as the latent heating keeps being present (at roughly similar magnitude), the divergent wind pattern keeps on being initiated at the location of the cell/updraft (e.g. [7, 8, 9]). Hence, the divergence signal sticks to the neighbourhood of the cumulonimbus cloud. As long as the box of integration then covers the precipitation region, but not a too large region beyond (where subsidence will compensate the updraft [5] and e.g. the added Figure 6 [1], previously part of the supplement), the estimate of the divergence will be accurate. In other words, by integrating over a region covering the precipitation cells, the divergence is more or less optimally integrated. Quantitative and qualitative analysis in our work (e.g. Figure 6 in the updated manuscript, and further analysis of the simulation dataset, not included in the manuscript) confirms this.

Regarding the later questions about compensation of neighbouring cells: this compensation occurs once the outflows (gravity waves) collide, over a larger integration volume with multiple convective cells in it. Therefore, if a volume containing for example an MCS contains multiple cells, effectively the outflows from the individual convective cells collide, but only once the gravity waves travel over the distance equal to the spacing of individual cells. The effect cannot occur instantaneously as an outflow forms, while the divergence signal forms instantaneously with the outflow at the location of a convective cell. In addition, as mentioned in the previous paragraph, the divergence signal will essentially stick to the cumulonimbus cloud's close proximity. Only the flow perturbation itself propagates away, as the divergence sticks to the place where the velocity gradient initially was: the surroundings of the cumulonimbus cloud.

**3    Technical comments**

The technical comments of referee 1 [2] have all been addressed in the updated manuscript.

**4 Referee 2: reply to RC2 [3]**

**4.1 High level comments**

*'The authors point out that their model did not adequately simulate stratiform precipitation'*

This message has been interpreted correctly by the reviewer, although the specific comment requires re-formulation in the eyes of the authors: adequate simulation of stratiform precipitation cannot be discerned from simulations that do not cover a realistic case, i.e. intercomparisons of simulations cannot be verified against a reality that never happens (in the case of idealised simulations). Of course it is possible to say that most MCS have a large stratiform region, but given conditions of our simulations may just not be favorable for that. We haven't investigated this and verification of the intercomparison of LES is beyond the scope of our work. Nevertheless, the results of our manuscript are indeed only representative for purely convective heating regimes, and this has been correctly interpreted by the reviewer.

*'The manuscript has 5 figures and at least 5 supplemental figures and numerous references to material in the supplemental material. This was done to the extent that it seemed that an entire portion of the analysis is done in the supplemental. For example, section 4.2 discusses the impact of physics perturbations and almost entirely references Figure 5. However, Figure 5 only shows data by storm type. There is no indication of how the relationship changes with momentum or low – level stratification. Thus, I don't think Figure 5 can be cited as proof of the arguments in this section. I suggest either figuring out how to add this information to the figures included in the manuscript or break this manuscript into two separate manuscripts.'*

We agree that there are many figures in and references to the supplementary material in the initial submit. We will reconsider the supplementary material part by part, adding parts of it to the main manuscript, possibly parts of it to the appendix or keeping it in the supplement. Thereby, it will be assured that the grounding information for the discussion and conclusion is available in the results section.

Nevertheless, it is incorrect to suggest that no inferences on the role of convective momentum transport or low-level stratification in simulations with perturbed momentum transport or low-level stratification can be made: all of the perturbed simulations are contained in the plot of Figure 5. As the relation between upper tropospheric divergent outflow intensity and latent heating rates are structured according to storm morphology - outflow dimensionality and convective organisation - the impact of those two factors on divergence patterns is clear. However, for a given storm type, the relationship between the divergent outflow and latent heating rate hardly differs. By comparing the integration

- over two time intervals that are available,

- for the various storm types,

- and for various strengths of momentum transport perturbations

It is deduced that no monotonic relation between momentum transport perturbations and upper tropospheric divergence can be found at a given latent heating rate for any storm type. Furthermore, no systematic deviations from the ensemble mean and background scatter could be identified for differential initial stratification. Most simulations with momentum transport perturbations and initial stratification perturbations deviate from the ensemble mean weakly and in non-systematic ways. On the other hand, adding extra symbol categories to Figure 5 to show that this can be inferred from Figure 5 makes that Figure too busy and distracts a reader from the main point. Therefore, we think that the most convenient solution is to add a copy of Figure 5 as appendix or tabulate its data in an appendix/supplementary file, to allow for the distinction of simulations with perturbed convective momentum transport and initial stratification. The various classes of momentum transport, storm types and time intervals are thereby accessible separately.

Furthermore, the note that Figure 5 includes all the physics perturbations has been added explicitly in Section 3.4.

As the effect of moment transport and initial low-level stratification is close to negligible and the information can be added by reproducing the plot Figure 5 with slightly different lay-out, it would definitely not be worth writing a separate manuscript on those experiments from the authors' point of view.

Regarding the materials that are maintained in the supplement, this is for the following reasons:

- Figure S1 is essentially a time integrated accumuluation of Figure 2, which presents instantaneous simulated radar imagery. Therefore, only one of the two is considered essential. Figure S1 is available in addition to the the highly interested readers just in addition to Figure 2.

- Similarly, Figure S2 provides information complementing Figure 3 of the main manuscript.

- Figure S3 is an extended version of Figure 4 in the main material, where the (seemingly) most and least variable subsets appear in Figure 4 and any intermediate of those in Figure S3 for the highly interested readers.

Furthermore, the main messages of the paper are supported in Section 3.4-3.5, 4 and 5. Of these sections, no supporting material appears in the appendices or supplement in the updated manuscript.

*'Line 71: It is mentioned that the manuscript the diabatic forcing and resulting flow will be tracked using the methodology from Baumgart et al. (2019). What exactly is that methodology? Baumgart is only referred to the Introduction and nowhere else in the manuscript.'*

The authors suspect that the initial version of the manuscript has caused some confusion about the relation to other works, in particular error growth works. No methods of Baumgart et al. (2019) have been used [10]. Nevertheless, part of the motivation for this research project was to investigate their hypothesis and possible relation of upper troposhperic divergent winds to the error growth in NWP.
There was no implication in the pre-print that a method of [10] was used, as the reviewer suggested. The only thing that was meant is that we aim to provide analysis and methods that could complement their method and could provide an extension of their methods in error growth studies. As such, we aimed to provide an outlook to using the methods of [10] and our methods [1] in one study. This was possibly formulated in a slightly suggestive way in the pre-print.
The paragraphs about the relation to error growth dynamics have been restructured. Furthermore, in a manuscript currently prepared for submission, investigation of variability of the divergent winds in NWP will be covered and the relations with error growth studies are stronger. That is beyond the scope of the current manuscript [1].
To reduce the probability of confused readers substantially, and remove slightly suggestive of relations to other papers, we have adjusted this part of the introduction accordingly.

*'How are model boundaries treated in the model? Are some of the wave patterns seen in the vertical velocity due to reflection off the edges of the model? I assume this is described in Groot and Tost (2022), but it is a very important detail related to these runs and should be mentioned here as well'*

The boundary conditions indeed match those of [4]. A description of boundary conditions, similar to that work, has been added in the model set-up subsection.

*'The abstract says nearly 100 ensemble members are created. The manuscript indicates several ways the simulations were perturbed, but I don't see how the ensemble and physics perturbations listed resulted in 100 ensemble members.'*

To clarify, the abstract states that about 100 large eddy simulations have been used for this study: this includes the simulations listed in Figure 1 ($3 \times (10 + 5 + 4 + 3) = 66$, numbers in brackets, Figure 1). Furthermore, there are 8 ensemble simulations of finite length squall lines, 6 theta-profile perturbations for the other three modes of convection and 2 extended domain simulations. This makes a total of 82 simulations. By adding all the numbers in brackets for this Figure, this will be clarified. The number has been changed to "over 80" in the updated manuscript.

*'Are the regions over which the diagnostics integrated over the same area/size? If they are different, how could averaging over different areas impact your results? Could some of the regime differences be attributed to the area difference'*

The areas selected for integration differ by under 10% and are in all cases about 7000 square kilometers. To be sure that the area selection has no effect, additional analysis has been done and some key findings are mentioned in the discussion of the manuscript: in Section 4.1. Furthermore, some extended analysis of the large domain simulations (not described) has been executed separately (before the initial submit of the manuscript) and confirmed the statements in the manuscript. In addition, the spatial distribution of divergence signals in the former Supplement - now in the revised version Figure 6 - shows where the divergence signals occur in terms of spatial extents. All of the analyses support the main message presented in the manuscript:
"An integration mask covering the convective cores and ending just outside of the area of precipitation accumulation leads to the detection of a large proportion of the divergent outflows. Little dilution from

convergence/inflow may occur if appropriate vertical levels are selected for vertical integration." [1]

Of course, analyses over a much larger area outside of the convective precipitation region would reduce the consequent amount of divergence detected, as it occurs over the area covered by the integration mask. Essentially, if the precipitation originating from convective systems falls in a restricted area as the storm-relative flow is weak, the set-up is suitable for our analysis. Given high shear and moderate shear profiles, this means that low-level and upper-level flow may be non-zero in a framework relative to the system. To overcome misdetection of divergence as a result of moving precipitation, the winds have been set such that the storm stays very close to the domain center throughout the simulation.

Hence, the short answer is that the main results can certainly not be attributed to area differences between the integration masks.

*'Figure 4 and its analysis is very confusing. When I looked at the figure several statements that were made in the manuscript did not seem consistent with the figure. For example: not everything developed intense convection within 30 mins (line 286) and not everything in the top and bottom row has strong convergence within 45 minutes (line 290). I also don't see the white line dipping down to 4 km (line 304). In general, I suggest referring to the type of storm in each panel not its row or column position. Please explain the white lines earlier. Additionally, is the data above 14 km really needed?'*

The statements are phrased slightly different in our manuscript, with some indications of the variability. It should be obvious that convective systems take some time to develop in large eddy simulations with initial conditions of large CAPE and strong forcing. That was also seen in Figure 2 of the pre-print. Therefore, mentioning that the first roughly 30 minutes are spent on initiation seems fair to the authors.

The Section and Figure 4 of the pre-print is supposed to illustrate what happens in terms of divergence and convergence. A general description on major differences between the array of simulations is presented. The word "about" was included to point at variations, but not that it is strictly 40-50 minutes or any specific variation. The authors agree that there is a lot of variation around the 45 minutes, and add a plus-minus 20 minutes statement in the revised version. However, after getting familiar with the main divergence and convergence of air in the simulations, the more important point of this section is the white isolines in the figure 4, and its discussion.

With regards to the white lines dropping to 4 km altitude, we refer to the bottom panel with the infinite-length squall lines after 60 minutes (55-65 minutes) of simulation time, where all ensemble members have a drop to below 5 km (but above 2.5 km).

Indication of storm types and sub-panel labelling are added.

The authors believe that limiting the panels to 14 km would be too low. However, it is indeed very obvious that 6 km higher in the stratosphere little to no divergence is supposed to occur. Just for the sake of the possibility of tropopause-overshooting, a marginal part of the stratosphere is better included. Key message has to be that it justifies the vertical mask for the next section.

*'Section 3.4: Talks a lot about changes in slope. It is very hard to visualize. Please create the regressions, show the values of the slope, and prove that they are significantly different using $R^2$.'*

This would of course be a valid method to quantify whether significant differences between two regimes exist. However, the exact value of the correlations would strongly be affected by which exact physics perturbations are applied, and how many of them are included. For example: it has been noted that -40% latent heating leads to a different depth of the convective system, and outflow at different levels. As a result, some -40% latent heating perturbations appear to be outlier points in the distribution of Figure 5. If we adjust the integration mask for these simulations to be better in line with the outflow altitude (as it is for nearly all other simulations), the outliers move towards the center of a given storm morphology (this has been mentioned in the discussion section, lines 422-424 in the pre-print). On the other hand, initial condition uncertainty tends to reduce scatter. The increased weighting of these simulations tends to increase the value $R^2$. Hence, correlation values for each type of storm would sensitively depend on the balance between exactly these simulations. Therefore, there is little added value provided by adding quantitative statements based on such analysis.

The authors agree that visualisation of the slopes of various categories may be helpful for the interpretation. Nevertheless, provided that statistical analysis of the correlation values and significance measures do not make sense statistically and methodologically, we would say that it's more confusing - it could suggest that statistical measures like correlation or significance would be useful information, while it is not in our eyes.

Solely by the structure of all simulations - ensemble members, those with perturbed physics and with different resolution or larger domains - it can be seen that the crosses and dots are nearly perfectly separated in the scatter. The crosses represent a nearly-3D regime and the dots a nearly-2D regime. This separation is

sufficient to conclude that there is a high degree of agreement with earlier works, like those based on the linear gravity wave model. The cited literature and explanations provide further evidence for the two limit regimes.

*'One of the main conclusions is that there whether the outflow is two dimensional or three dimensional is determined by storm morphology. There was no figure that explicitly showed it. If something is included in the abstract and conclusions sections, it needs to be proven with a figure in the manuscript – not in the supplemental'*

As the reviewer's comment suggests, the relation of the two regimes to 'storm morphology' or storm type was indeed clarified in further detail in the supplementary material (in the pre-print version). In the new version, the main clarifying figure appears in the main text (Figure 6 there) and visualisation is only included for the finite-length squall line, as it has regions that resemble a 2D outflow and regions resembling a 3D outflow. Hence, the patterns could be extrapolated to the infinite length squall line and supercell/multicell - two components of the outflow are similarly important (supercell/multicell) or negligible (infinite length squall line).

Further content of the comment seems to overlap with comment 1 by the same reviewer [3], and we assume that the reconsideration mentioned as a reply to that comment (mostly) solves the concerns of the reviewer with regards to this point.

**4.2 Technical comments**

*'1. I found the sentence structure in this manuscript to be convoluted. The sentences are very long and use numerous commas. While present throughout the manuscript, the introduction was particularly difficult to read. I encourage the authors to break their sentences into small, more succinct sentences. Here are just a few examples: a. Lines 46 – 51, Line 262 – 263,*

*2. I was a bit confused by the organization of the introduction. Lines 1 – 44 and 57 – 74 seem to be background information. However, lines 45 – 56 state the objectives of the paper. This organization felt jumpy and caused me some confusion. I suggestion combining lines 1 – 44 and 57 – 74 and then stating the objectives and outline of the paper.*

*3. I suggest using the oxford comma throughout the manuscript. There are numerous commas throughout the paper. However, without the use of the oxford comma I found myself getting confused as whether something was a list or an aside.'*

The authors did not have the impression that there are so many long convoluted sentences throughout the manuscript, only in some sections. Nevertheless, the usage of commas and the structure of sentences, as well as the last part of the introduction is improved in terms of clarity for the re-submission, especially in some sections (e.g. end of 1 and 3.4). The authors hope that the improved manuscript is unambiguous and straightforward to read and interpret.

Specifically on the introduction: lines 1-44 explain the context of why the hypothesis that convective organisation can affect upper tropospheric divergent outflows is studied, and which material is very relevant. Then, the main objectives are stated indeed in the following 12-15 lines, later on specifying the methods. This is followed by some relevant implications to weather and climate modelling that make the dependence of the outflows on convective organisation interesting for theoretical knowledge and more practical applications are discussed. However, this later part relevant for weather and climate modelling is now adjusted and moved to a location above the main objectives, which hopefully is more convenient to the readers.

*'5. Lines 24 – 26: I disagree with this statement. It has been shown that, overtime, stratiform regions can create just as much or more latent heat release than convective regions. Stratiform regions have lower precipitation rate than convective regions. This is especially true for mesoscale scale convective systems, which often include squall lines. Also, what does "(intuitively)" mean, imply here?'*

The statement is meant to say that flow response (e.g. as a consequence of gravity wave activity) scales with the strength of latent heat release in the column, which is an intuitive finding by [6]. The scaling both applies to stratiform and convective parts of a system [11, 12, 6]. It is not meant to imply anything about the contrast between stratiform parts and convective cores of MCS. The latter of these obviously have the higher area averaged precipitation intensity locally, but averaged over say 10.000+ square kilometers and several hours to almost a day, or even longer time scales, the heating in the column may often be similarly strong between the two (as it may be inferred from for example [13]).

The statement should be interpreted as a preparation for the linearised gravity wave model that explains the "intuitively" and the relation to latent heating in the following paragraph, a relation which is meant to

be emphasised by the repetition in the first words of the next paragraph. Furthermore, increase of average vertical motion with precipitation rates are intuitive, and stronger vertical motion intuitively perturbs the horizontal winds more strongly.

*'10. Line 186: Why was the simulated extended to 160 minutes when data was analyzed only out to 120 hours?'*

Extending the integration time at a large domain was sensible, because the domains were larger and the flow effects escape the domain over a longer time scale than for the standard set-up. For the simulation intercomparison essentially no comparison could be made to the other simulations over the last 40 minutes. In spite of that, some additional examination of diagnostics has been done in the large domain simulations that has been of benefit to the discussion section.

*'13. When referring to higher latent heating rates please be specific. What I deem high maybe different than you.'*

From the context and dataset that is provided, it may be straightforward to infer where the separation between high and low latent heating rates is in our analysis. However, we will label a value to the separation between the high latent heating rates and the low ones for convenience (roughly 3000 Watts per square meter latent heating, about 4 mm/h precipitation rate).

Additional technical comments have all been addressed in the revised manuscript by implementing specific changes to solve the issues.

**References**

[1] Edward Groot and Holger Tost. Evolution of squall line variability and error growth in an ensemble of les. *EGUsphere*, 2022:1–34, 2022.

[2] Referee comment 1 on "divergent convective outflow in large eddy simulations", 2023.

[3] Referee comment 2 on "divergent convective outflow in large eddy simulations", 2023.

[4] E. Groot and H. Tost. Evolution of squall line variability and error growth in an ensemble of large eddy simulations. *Atmospheric Physics and Chemistry.*, 2023.

[5] Christopher S. Bretherton and Piotr K. Smolarkiewicz. Gravity waves, compensating subsidence and detrainment around cumulus clouds. *Journal of Atmospheric Sciences*, 46(6):740 – 759, 1989.

[6] Melville E. Nicholls, Roger A. Pielke, and William R. Cotton. Thermally forced gravity waves in an atmosphere at rest. *Journal of Atmospheric Sciences*, 48(16):1869 – 1884, 1991.

[7] Rebecca D. Adams-Selin and Richard H. Johnson. Examination of gravity waves associated with the 13 march 2003 bow echo. *Monthly Weather Review*, 141(11):3735 – 3756, 2013.

[8] Rebecca D. Adams-Selin. Impact of convectively generated low-frequency gravity waves on evolution of mesoscale convective systems. *Journal of the Atmospheric Sciences*, 77(10):3441 – 3460, 2020.

[9] Rebecca D. Adams-Selin. Sensitivity of mcs low-frequency gravity waves to microphysical variations. *Journal of the Atmospheric Sciences*, 77(10):3461 – 3477, 2020.

[10] Marlene Baumgart, Paolo Ghinassi, Volkmar Wirth, Tobias Selz, George C. Craig, and Michael Riemer. Quantitative view on the processes governing the upscale error growth up to the planetary scale using a stochastic convection scheme. *Monthly Weather Review*, 147(5):1713–1731, May 2019.

[11] Robert A. Houze. Mesoscale convective systems. *Reviews of Geophysics*, 42(4), 2004.

[12] Brian E. Mapes. Gregarious tropical convection. *Journal of Atmospheric Sciences*, 50(13):2026 – 2037, 1993.

[13] Courtney Schumacher, Robert A. Houze, and Ian Kraucunas. The tropical dynamical response to latent heating estimates derived from the trmm precipitation radar. *Journal of the Atmospheric Sciences*, 61(12):1341 – 1358, 2004.

---

## Referee Report (RR1)

**Summary:**

This study analyzes the relationship between upper – level divergence and latent heat/precipitation using numerous large LES simulations of supercells, multicellular convection, or squall lines. There simulations have little stratiform precipitation, so this study mainly comments about divergence in convective regions. It is found that the amount of upper – level divergence is primarily dependent on the amount of latent heat/precipitation and storm morphology. Additionally, storm structure tends to dictate whether the outflow is 2D or 3D as the storm initially forms. Convection without surrounding cells, such as the super cell, multi cell, and the ends of the finite squall line, start with 3-dimensional outflows. Convection consistently has nearby/surrounding convection, such as the infinite squall line and center of the finite squall line, tend to have 2-dimensional outflows. At later times, storms tend to be a hybrid between 2 and 3 dimensions.

First, I want to complement the authors for their hard work in reviewing this manuscript. I found it much improved, especially the second half of the manuscript. However, I still found a few portions unclear, and I still disagree with a few analyses. Thus, I suggest return for minor revisions.

**Comments:**

1. Introduction: lines 51 – 78: I found this section confusing and hard to follow. The first paragraph in this section describes the potential impact of this study, but the details about what exactly is done in this study is not really discussed until the second and third paragraph. And when it is discussed, it is scattered throughout the paragraphs. I suggest reworking these paragraphs. Start with clear statements of what exactly will be done and what is the objective of this work. Then discuss possible implications of this work. The discussion of the implications of this work could also be fully or partially moved to the conclusions section.

2. Section 3.2:
   a. Line 281: What are you comparing the supercell to? The multicell case? Please specify in the text.
   b. Line 284: Are the boxes in the supercell and multicell only similar in size or are the boxes the same size for all the cases? How does this impact the results since it is later mentioned that results are sensitive to whether just the convective cells are included or if some region outside of the convective region is included?
   c. Lines 285 – 288: Can you provide the x,y location of these regions influenced by the boundary conditions? I am thinking it is the vertical bands are +/- 50-40 x. If that is true, could the slightly arched bands in the finite squall line between +/- 40-60 x and +/- 20 y also be boundary artifacts?

3. Section 3.3:
   a. Line 304 – 310: I got confused in this section. Can you review it and make sure the text is referring to finite or infinite correctly as well as review the panel labels.
   b. In the text, instead of simply stating that the spread is larger based on the spaghetti lines, could you provide a standard deviation or variance of the ensembles to quantify and be more exact?
   c. Figure 4: I understand the importance of the white lines between 4-8 km. However, the white lines above 15 km are never discussed. Those lines tend to make the figures busier and more confusing. Could you remove those upper lines or add text that specifically explains why including them is important?

4. Section 3.4:
   a. Lines 344 – 348: Are the slopes here calculated only using the first and last point in this region? This is not an adequate approach; a linear regression must be

made using all points in this region. I expect that the slopes will be more similar if a true linear regression is conducted.

    b. Line 349: Is this supposed to say "finite"? Please review this entire paragraph to make sure it is referring to the correct case.

    c. Lines 350 – 355: Please define exactly what is the end point region? Is this comment based solely on the last point in finite distribution and, if so, why is it ok to use a possible outlier as the end point? In my interpretation the end point region would extend from 4000-6000 latent heating flux, and, in that case, the mass divergences are not similar.

    d. Line 360 – 383: The introduction of 2D and 3D seemed very abrupt and out of place to me. I had no idea how the 2D and 3D idea came about and how it related to these cases. Lines 373-380 nicely describe how each case fits the 2D and 3D models. I suggest moving these lines up and then stating lines 360-370.

    e. Line 391: Least affected by what? Please be specific.

5. Section 3.5:

    a. Lines 410 – 412:

        i. I don't see the v-component in the southern end initially. I suggest not mentioning.

        ii. Is there any implication for the northern v divergence magnitude being similar to the v convergence in the center of the line?

        iii. It may be worth noting that the u and v divergence magnitudes at the northern end are similar in magnitude to help strengthen the idea that it is 3D.

---

## Author Response (AR2)

**Replies to new Referee Comments on "Evolution of squall line variability and error growth in an ensemble of LES " [1]**

Edward Groot, Holger Tost

April 14, 2023

**1 General**

We thank the referee for filtering out and formulating their pointwise summary and feedback [2].

**2 Comments**

*'Introduction: lines 51 – 78: I found this section confusing and hard to follow. The first paragraph in this section describes the potential impact of this study, but the details about what exactly is done in this study is not really discussed until the second and third paragraph. And when it is discussed, it is scattered throughout the paragraphs. I suggest reworking these paragraphs. Start with clear statements of what exactly will be done and what is the objective of this work. Then discuss possible implications of this work. The discussion of the implications of this work could also be fully or partially moved to the conclusions section. '*

In case the paragraphs on the relation to predictability studies (e.g. [3, 4]) are not reader-friendly yet, we have considered moving those to the implications section the best option. Additionally, we have made some improvements and restructured the other paragraphs to something that should be more reader-friendly too.

*'2. Section 3.2: a. Line 281: What are you comparing the supercell to? The multicell case? Please specify in the text. b. Line 284: Are the boxes in the supercell and multicell only similar in size or are the boxes the same size for all the cases? How does this impact the results since it is later mentioned that results are sensitive to whether just the convective cells are included or if some region outside of the convective region is included? c. Lines 285 – 288: Can you provide the x,y location of these regions influenced by the boundary conditions? I am thinking it is the vertical bands are +/- 50-40 x. If that is true, could the slightly arched bands in the finite squall line between +/- 40-60 x and +/- 20 y also be boundary artifacts? 3. Section 3.3: a. Line 304 – 310: I got confused in this section. Can you review it and make sure the text is referring to finite or infinite correctly as well as review the panel labels. b. In the text, instead of simply stating that the spread is larger based on the spaghetti lines, could you provide a standard deviation or variance of the ensembles to quantify and be more exact? c. Figure 4: I understand the importance of the white lines between 4-8 km. However, the white lines above 15 km are never discussed. Those lines tend to make the figures busier and more confusing. Could you remove those upper lines or add text. 4. Section 3.4: b. Line 349: Is this supposed to say "finite"? Please review this entire paragraph to make sure it is referring to the correct case. c. Lines 350 – 355: Please define exactly what is the end point region? Is this comment based solely on the last point in finite distribution and, if so, why is it ok to use a possible outlier as the end point? In my interpretation the end point region would extend from 4000-6000 latent heating flux, and, in that case, the mass divergences are not similar. e. Line 391: Least affected by what? Please be specific.'*

To all these points further specification of details has been added, to complete and clarify the information requested by the reviewer.

*'4. Section 3.4: a. Lines 344 – 348: Are the slopes here calculated only using the first and last point in this region? This is not an adequate approach; a linear regression must be made using all points in this region. I expect that the slopes will be more similar if a true linear regression is conducted.'*

Here, a semantical issue has been resolved. The implicit (silent...) assumption was made by the authors, that each point in Figure 5 [1] is connected to a point at (0, 0) in the Figure. This has not been mentioned

and was confusing.

As detailed on in the previous round of review (see pages 5-6 of that reviewer reply), no regression analysis could be done. Instead, each record in Figure 5 [1] is supposed to represent an individual mass divergence-latent heating ratio; the collection of these points behave in a way as explained in the work. In the updated manuscript we refer to this ratio between mass divergence and net latent heating, also defined as normalised latent heating, to circumvent this purely semantic issue: the suggestion of regression analysis and fitting procedure by specifically mentioning the word "slope(s)".

*'4. Section 3.4: d. Line 360 – 383: The introduction of 2D and 3D seemed very abrupt and out of place to me. I had no idea how the 2D and 3D idea came about and how it related to these cases. Lines 373-380 nicely describe how each case fits the 2D and 3D models. I suggest moving these lines up and then stating lines 360-370. '*

We accommodated these changes and made additional minor changes to improve the specific part of the manuscript.

*'5. Section 3.5: a. Lines 410 – 412: i. I don't see the v-component in the southern end initially. I suggest not mentioning. ii. Is there any implication for the northern v divergence magnitude being similar to the v convergence in the center of the line? iii. It may be worth noting that the u and v divergence magnitudes at the northern end are similar in magnitude to help strengthen the idea that it is'*

The first point (i) of the reviewer is partly correct. Nevertheless, when one zooms in at the mentioned Figure 6 [1], an area that exceeds the threshold of the first, lowest, divergence threshold is found. However, this area is very small (probably hard to spot, even when zooming). Effectively, the notion that divergence is negligible suffices indeed. Therefore, the updated manuscript accommodates suggestion that it is not needed to be mentioned.

With regards to the second sub-bullet(ii), this is difficult to answer, as in this region the convergence is generated by the divergence collision of multiple individual ascents, for which a separation is no longer possible. Therefore, it cannot be answered directly; however, the overall meridional divergence in the infinite length squall line is similarly close to zero as it initially is in the centre of the finite length squall line (see Fig. 6b of the manuscript.) We agree with the statement of sub-bullet(iii), which is also confirmed by Fig. 6 .

**References**

[1] Edward Groot and Holger Tost. Evolution of squall line variability and error growth in an ensemble of les. *EGUsphere*, 2022:1–34, 2022.

[2] Anonymous referee comment on "divergent convective outflow in large eddy simulations", submitted on 17 mar 2023.

[3] Mark J Rodwell, Linus Magnusson, Peter Bauer, Peter Bechtold, Massimo Bonavita, Carla Cardinali, Michail Diamantakis, Paul Earnshaw, Antonio Garcia-Mendez, Lars Isaksen, Erland Källén, Daniel Klocke, Philippe Lopez, Tony McNally, Anders Persson, Fernando Prates, and Nils Wedi. Characteristics of occasional poor medium-range weather forecasts for europe. *Bulletin of the American Meteorological Society*, 94(9):1393–1405, September 2013.

[4] Marlene Baumgart, Paolo Ghinassi, Volkmar Wirth, Tobias Selz, George C. Craig, and Michael Riemer. Quantitative view on the processes governing the upscale error growth up to the planetary scale using a stochastic convection scheme. *Monthly Weather Review*, 147(5):1713–1731, May 2019.